



# Historic hydrological droughts 1891-2015: systematic characterisation for a diverse set of catchments across the UK

Lucy J. Barker[1], Jamie Hannaford[1, 2], Simon Parry[1], Katie A. Smith[1], Maliko Tanguy[1], Christel Prudhomme[3, 1, 4]

[1] Centre for Ecology & Hydrology, Wallingford, UK
[2] Irish Climate Analysis and Research UnitS (ICARUS), Maynooth University, Ireland
[3] European Centre for Median-range Weather Forecasts, Reading, UK
[4] Loughborough University, Loughborough, UK

*Correspondence to*: Lucy J. Barker (lucybar@ceh.ac.uk)

**Abstract.** Hydrological droughts occur in all climate zones and can have severe impacts on society and the environment. Understanding historical drought occurrence and quantifying severity is crucial for underpinning drought risk assessments and the developing drought management plans. However, hydrometric records are often short and capture only a limited range of variability. The UK is no exception: numerous severe droughts over the past 50 years have been well captured by observations from a dense hydrometric network. However, a lack of long-term observations means that our understanding of drought events in the early 20th century and late 19th century is limited. Here we take advantage of new reconstructed flow series for 1891 to 2015 to identify and characterise historic hydrological droughts for 108 catchments across the UK using the Standardised Streamflow Index. The identified events are ranked according to four event characteristics (duration, accumulated deficit, mean deficit and maximum intensity), and their severity reviewed in the context of events of the recent past (i.e. the last 50 years). This study represents the first national scale assessment and ranking of hydrological droughts. Whilst known major drought events were identified, we also shed light on events which were regionally important such as those in 1921 and 1984 (which were important in the south-east and north-west of the UK, respectively). Events which have been poorly documented such as those of the 1940s in the post-war years, or the early 1970s (prior to the landmark 1975-1976 event), were found to be important in terms of their spatial coverage and severity. This improved knowledge of historic events can support improved long-term water resources planning approaches. Given the universal importance of historical drought appraisal, our systematic approach to historical drought assessment provides a methodology that could be applied in other settings internationally.

## 1 Introduction

In all climate zones, droughts are a major natural hazard and can threaten water supplies and trigger severe societal and environmental consequences (e.g. Bachmair et al., 2016). Proactive drought risk assessment and planning are essential cornerstones of efforts to manage the impacts of droughts in many countries



(Wilhite et al., 2000). Such activities rely on an understanding of the likelihood of droughts of a given severity, in addition to information on vulnerability of supply infrastructure, populations, ecosystems etc. The likelihood of drought occurrence is contingent on an understanding of past hydrometeorological variability, which in itself depends on long historical records of observational data (of rainfall, evapotranspiration, river flows, groundwater etc.). While water resources and drought planning efforts have evolved over the last three decades to incorporate climate model-based assessments of future climate variability under anthropogenic warming scenarios (Brown et al., 2015), the inherent uncertainties in these simulations mean that historical records are still of fundamental importance in drought planning – as well as providing the data to corroborate modelling projections and provide a baseline against which future changes can be assessed.

While the UK is a humid country, it has periodically suffered from severe droughts which have caused major water shortages and subsequent impacts. Parts of the UK are water stressed owing to a delicate balance between supply and demand – notably, in some of the drier areas of the south and east where some of the greatest concentrations of population live alongside intensive agriculture and commerce (Environment Agency and Natural Resources Wales, 2013). Consecutive dry winters pose a particular threat in these areas where groundwater makes up a large proportion of public water supply, as was demonstrated in the recent droughts of 2004-2006 and 2010-2012 (e.g. Parry et al., 2016). These reserves are reliant on recharge over the winter months to replenish supplies. In the wetter north and west, droughts may, intuitively, not be regarded as a major issue, but natural catchment storage is limited and even relatively short periods of warm, dry weather can cause significant risks to water supply (as occurred in summer 2018, e.g. Barker et al. (2018a)).

As with other regions, there remain large uncertainties in hydrological projections for the UK (Watts et al., 2015) but the general expectation of increased evaporative demand under anthropogenic warming is expected to trigger drying, particularly in summer months (Prudhomme et al., 2011). Moreover, greater climatic variability could mean an increase in persistent blocking episodes and multi-year droughts, which are the greatest challenges in the most vulnerable areas of south-east England. These factors have led the UK Climate Change Risk Assessment to identify water scarcity as a major risk for the UK (Adaptation Sub-Committee, 2016). Even without climate change impacts, demographic and economic changes are expected to significantly influence future water demands (e.g. Water UK, 2016; National Infrastructure Commission, 2018), and the need to ensure favourable conditions for aquatic ecology places a constraint on future water availability (Environment Agency, 2009).

There is therefore, a pressing need for improved tools for drought risk assessment, the development of which is contingent on a proper quantification of past occurrence of droughts in the UK. Droughts are a complex hazard and it is crucial to quantify understand not just a peak intensity, but duration and spatial extent, all of which are interdependent and different for individual events (Van Loon et al., 2016). Given



the usually large spatial footprint and long timescales of drought, it is also challenging to define drought episodes as self-contained events (and as a result, their onset, termination, seasonality etc.), underlining the importance of consistent, quantitative methods for drought identification.

Knowledge of past droughts is crucial for supply system planning. In the UK, as in many countries, water resources management plans and droughts plans have long relied on a 'drought of record', i.e. using the worst observed historic drought to test the resilience of supply systems (Environment Agency, 2015). More recently, there has been a shift towards stochastic approaches to test the resilience of systems to droughts that are worse than those observed in the recent past (Anderton et al., 2015; Water UK, 2016). These approaches recognise the need to go beyond the envelope of past variability, not just in the context of climate change but given short observational records, wherein it may be expected that 'record breaking' events will occur due to chance alone (as has been described for flooding events, for example: Thompson et al. (2017); Kjeldsen and Prosdocimi (2018)). However, these stochastic approaches still require benchmarking against historic data, and where longer historic data are available the increased sample size increases the confidence in synthetic events generated using historical training data.

The occurrence of droughts in recent decades is well understood in the UK, with most events since 1976 having been documented extensively by the National Hydrological Monitoring Programme (http://nrfa.ceh.ac.uk/nhmp). However, our understanding of hydrological drought occurrence is grounded in the period since 1960 when most UK river flow records commenced. Only a handful of hydrometric records extend back to the early 20th century meaning there are few observations on which to base systematic, national scale assessments of drought severity. In a seminal study, Marsh et al. (2007) (see also: Cole and Marsh, 2005) synthesised a range of datasets, to provide an assessment of historic droughts in England and Wales between 1800 and 2006, identifying nine 'major' episodes of which only four are in the well-gauged period of the last five decades. Whilst the study rightly encouraged a longer view, the approach to drought characterisation was qualitative and relied on a small number of long rain gauge and groundwater records, as well as documentary sources.

The British Isles is blessed with plenty of long rainfall series. This has allowed quantitative identification of meteorological droughts back to the 17th century, e.g. Todd et al. (2013). Similarly, in Ireland, Noone et al. (2017) developed a drought catalogue back to the 18th century using long rainfall records. But in the context of drought, it is not sufficient to quantify meteorological deficits alone. As some of the most severe drought impacts on society and the environment result from hydrological drought (i.e. deficits in river flows and groundwater levels), and given that the propagation from meteorological to hydrological drought is highly non-linear (e.g. Barker et al., 2016), assessments based solely on meteorology can be misleading (Van Lanen et al., 2013).

Given the lack of long river flow records, such long rainfall records can be used to reconstructed river flow data to extend our knowledge of past variability. The most notable existing example is the work of



the Climate Research Unit which has allowed a window into past droughts back to 1865 (Jones, 1984; Jones and Lister, 1998; Jones et al., 2006). However these studies reconstructed river flows for just 15 catchments across England and Wales, were based on empirical rainfall-runoff relationships and made a number of simplifying assumptions such as the use of constant potential evapotranspiration through time

(which plays an important role in discharge generating processes, particularly in the mid-latitudes). There have been some efforts to reconstruct hydrological droughts on a regional scale, the results of which have been run through water supply system models: e.g. for East Anglia (Spraggs et al., 2015) and the Midlands (Lennard et al., 2016). At the national scale, an assessment of past hydrological droughts was recently undertaken by Rudd et al. (2017) which benchmarked a national gridded hydrological model against the

droughts identified in Marsh et al. (2007) aggregated over river basin regions, but did not assess the relative severity or spatio-temporal dynamics of historical episodes. Internationally, although many studies have identified and described historic periods of meteorological drought (e.g. Noone et al., 2017; Pfister et al., 2006; Spinoni et al., 2015), there have been few efforts to reconstruct historic hydrological droughts at the broad, national scale. An exception is Caillouet et al. (2017) which used rainfall-runoff

model based reconstructions for 662 catchments across France.

The aim of this study is to provide the first comprehensive assessment of historic hydrological droughts at the UK scale; providing an up-to-date, objective and quantitative assessment of the severity of major droughts, extending the work of Marsh et al. (2007) (hereafter, for brevity referred to as MCW2007). However, unlike MCW2007, this study focuses on hydrological, specifically river flow, drought. River

flow integrates upstream processes combining the effects of climate and the physical catchment properties and is therefore good indicator of water availability. This study is part of 'Historic Droughts' (historicdroughts.ceh.ac.uk), a multidisciplinary project, which aims to understand past drought from a range of perspectives, with a hydrometeorological assessment at its foundation.

This paper:

• Presents timelines of historic reconstructed droughts for over 100 near-natural catchments,

• Characterises the severity of these past drought events – in terms of duration, accumulated deficit, mean deficit and maximum intensity,

• Ranks historic droughts according to these drought event characteristics and assesses how relative rankings vary by geography and the ranked drought characteristic, and

• Provides a fuller description of the evolution and characteristics of major, nationally important droughts from the pre-1961 period.





## 2 Data and Methods

### 2.1 Data

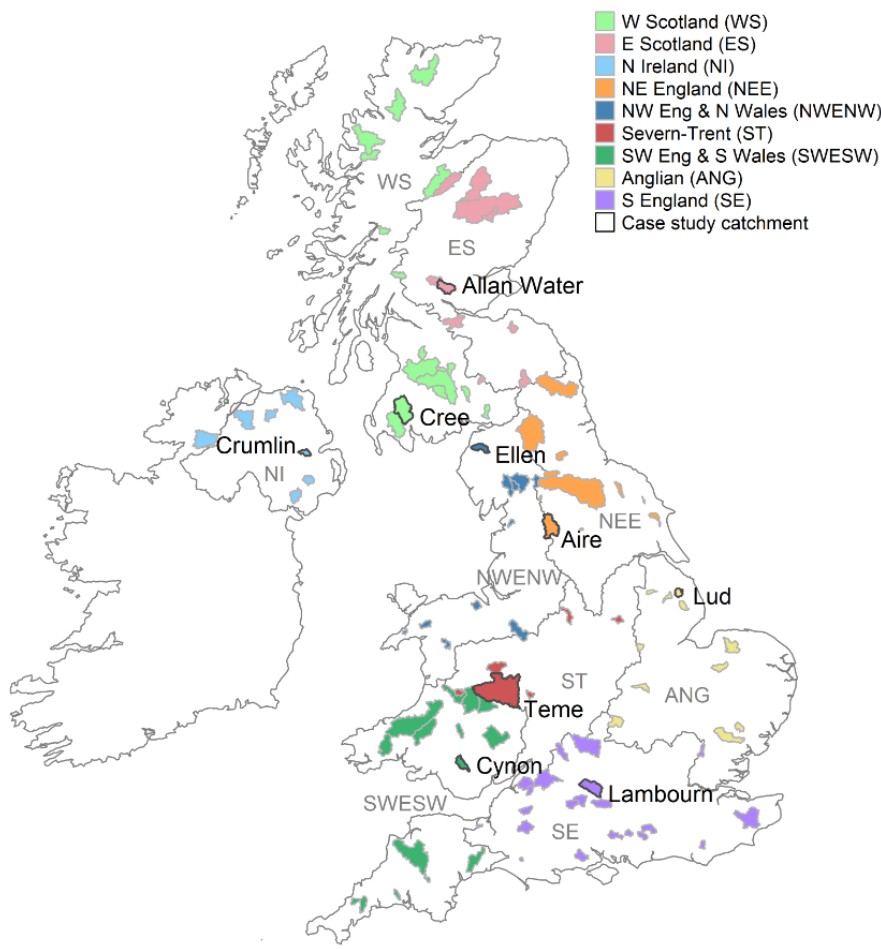

**Figure 1 The 108 low flow Benchmark Network (LFBN) catchments used in this study which are included in the river flow reconstructions of Smith et al. (2018), highlighting nine selected case study catchments. Catchments are coloured by the hydroclimate regions of Harrigan et al. (2018).**

This study makes use of a comprehensive new dataset of reconstructed river flows from 1891 to 2015 for 303 diverse catchments across the UK (Smith et al., 2018). These reconstructed daily river flows were derived from a hydrological model, driven by newly rescued and digitised meteorological data (Legg and McCarthy, in preparation). The hydrological model required daily rainfall and potential evapotranspiration as inputs, the latter of which was calculated using the newly recovered temperature data (PET; Tanguy et al., 2018). The hydrological model employed was the GR4J daily lumped rainfall-runoff model (Perrin et al., 2003), implemented using the 'airGR' R package version 1.0.2 (Coron et al., 2017) as described by Smith et al. (2018) and Smith et al. (2019). GR4J was calibrated using a Latin Hypercube Sampling (LHS) technique to ensure each parameter was sampled in an efficient manner,



producing 500,000 model parameter sets for each catchment. The 500,000 model results were then analysed and ranked using six evaluation metrics which assessed model performance across the flow regime but included drought and low flow specific metrics (Smith et al., 2019). From the 500,000 model runs for each catchment, Smith et al. (2018) identified the best performing model run – referred to as

LHS1. Smith et al 2019 also assessed the model parameter uncertainty for the top 500 model runs, but here, the LHS1 dataset was used to investigate historic hydrological droughts. LHS1 was selected due to the computational demand of the distribution fitting associated with the derivation of the Standardised Streamflow Index (see below).

In this study, we used a subset of the 303 catchments modelled by (Smith et al., 2018), i.e. stations from

the National River Flow Archive's (NRFA) UK Benchmark Network (Harrigan et al., 2017), in particular those stations suitable for low flows, hereafter referred to as the low flow Benchmark Network (LFBN). The NRFA's UK Benchmark Network provides a network of gauging stations monitoring near-natural catchments, with limited net artificial influences on flows (Harrigan et al., 2017). The use of near-natural catchments enables the hydro-climatic signal to be separated from confounding impacts (such as human

modifications to catchments or influences on flows); especially vital given that human impacts are not explicitly accounted for in the modelling approach used (Smith et al., 2019).

The 303 UK catchments reconstructed by Smith et al. (2018; which include the LFBN) performed well in model validation steps which included assessing model performance for a range of metrics which summarised how well the model reproduced discharge across the flow regime, as well as testing the skill

of the model for low flows and drought specifically (Smith et al., 2019). Here, where model evaluation criteria fell below the highest thresholds of the model performance metrics for the LFBN (for more detail, see Smith et al., 2019), catchments were removed. The LFBN generally performed well, and only seven catchments were excluded from the full LFBN (115 catchments), resulting in 108 catchments appropriate for this study (shown in Figure 1). To provide some geographic context to figures, hydroclimate regions

of the UK described in Harrigan et al. (2018) (shown in Figure 1) and were used in the description of the results. A set of nine case study catchments was chosen from the LFBN (one per hydroclimate region, shown in Figure 1) representing a range of catchment types and geographies across the UK, enabling catchment-scale results to be shown. The following case study catchments were selected (the hydroclimate region is given in brackets): Cree (WS), Allan Water (ES), Crumlin (NI), Aire (NEE), Ellen

(NWENW), Teme (ST), Cynon (SWESW), Lud (ANG) and Lambourn (SE).

## 2.2 Drought indicators

The Standardised Streamflow Index (SSI; Vicente-Serrano et al., 2011) was calculated for each LFBN catchment using reconstructed flow data (Smith et al., 2018) for the period 1891-2015 (Barker et al., 2018b). The standardisation of the reconstructed streamflow allowed consistent comparison over both



time and space and provided a measure of drought severity – crucial characteristics for a quantitative and rigorous assessment of drought event characteristics over time and space.

Daily mean river flow reconstructions were aggregated to mean monthly flows for each catchment in the LFBN. The SSI was then calculated for 3 and 12 end-month accumulation periods. The 12-month

accumulation period (SSI-12) gives a summary of long-term annual (multi-season) deficits likely to have greater impact on water resources (whether groundwater or multi-season reservoirs). The 3-month accumulation period (SSI-3) characterises short-term seasonal river flow deficits and impacts on smaller, single season reservoirs. The SSI was calculated using the Tweedie distribution, which has been found to have the best fit for observed river flow data for UK Benchmark catchments (Svensson et al., 2017). Due

to the uncertainties associated with extreme SSI values (e.g. Stagge et al., 2015), values were limited to the range -5 to 5.

Although the daily river flow reconstructions had no missing data, there were five individual months to which a Tweedie distribution could not be fitted and so did not have an SSI value. All five months of missing data appear in the SSI-3 series for four catchments and equate to 0.0015% of available data for

the 108 catchments and two accumulation periods for the period 1891-2015 (see Table S1 for details). Four of these missing data points occurred during periods of positive SSI (i.e. above normal flows) and so did not affect the identification of drought events (see Section 2.3). The last missing data point occurred in December 1921 for the Great Stour at Horton (South East England); and was infilled with the SSI value of preceding and subsequent months (both -5).

**2.3 Drought event extraction and characteristics**

Drought events were defined as months with consecutively negative SSI values with at least one month in the negative series reaching a threshold of -1.5 (equating to 'severe' drought; Barker et al., 2016). For each extracted event, the following characteristics were calculated (after Noone et al., 2017, see Figure 2):

• Duration (number of months),
   • Accumulated deficit (sum of SSI values across the event duration),
   • Mean deficit (accumulated deficit divided by duration), and
   • Maximum intensity (the minimum SSI value during the event).

Due to the seasonal focus of the study, events with a duration of less than three months were removed.

As the accumulated deficit and mean deficit are derived from the SSI, they represent relative deficits, not absolute flow deficits (for example, as mm or a volume)

The extracted events were ranked by each event characteristic (i.e. duration, accumulated deficit, mean deficit and maximum intensity) and the top 10 events for each characteristic and accumulation period



were identified. When ranking by duration, tied events were also sorted by the accumulated deficit so the longest events with the lowest (i.e. most negative) accumulated deficit ranked highest.

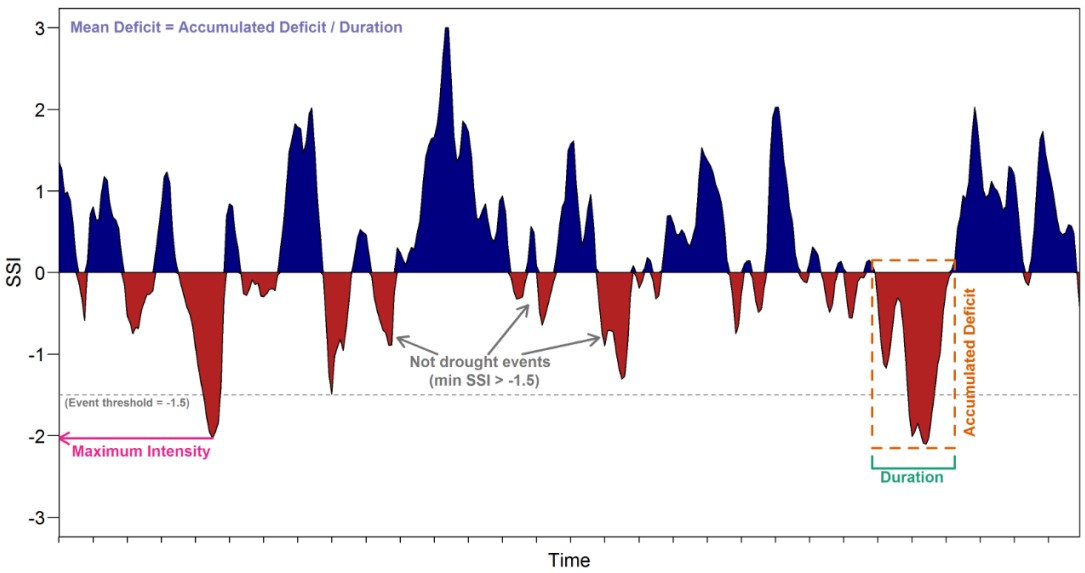

**Figure 2 Conceptual diagram illustrating drought event identification and characteristics (N.B. x-axis ticks represent years, but SSI data are on a monthly time-step).**

All extracted drought events were compared to events identified as 'major' droughts in England and Wales by MCW2007, or documented as such by the National Hydrological Monitoring Programme and UK Met Office (2004-2006 and 2010-2012 (e.g. Marsh (2007); and Kendon et al. (2013), respectively). Using information from their relevant publication, the start and end months were identified for each event (listed in Table 1). Where the extracted events (from the reconstructed SSI series) overlapped with the major event periods given in Table 1, the extracted event was assigned to the corresponding 'major' drought. Extracted events which did not overlap with these known drought periods, they were classified as 'other events' and were assessed in more detail.

**Table 1 Major droughts and their start/end dates as identified by MCW2007, asterisk denotes events not listed by MCW2007, but were significant events reported by the National Hydrological Monitoring Programme.**

| 'Major' Droughts | Start month | End month |
|---|---|---|
| ('The Long Drought') 1890-1910 | 1890-Jan | 1910-Dec |
| 1921-1922 | 1920-Sep | 1922-Mar |
| 1933-1934 | 1932-Sep | 1934-Nov |
| 1959 | 1959-Feb | 1959-Nov |
| 1976 | 1975-May | 1976-Aug |
| 1990-1992 | 1990-Mar | 1992-Aug |
| 1995-1997 | 1995-Mar | 1997-Aug |



| 'Major' Droughts | Start month | End month |
|---|---|---|
| *2004-2006 | 2004-Feb | 2006-Oct |
| *2010-2012 | 2010-Jan | 2012-Mar |

## 3 Results

### 3.1 Timelines of historic reconstructed drought

Figure 3 provides a national scale assessment of drought occurrence, showing the SSI-12 in the form of a heatmap, with catchments orientated broadly north to south. Spatially coherent phases of below normal flows (referred to as low flows throughout this section) can be identified in Figure 3, with particularly intense periods of low flows in the mid-1930s and 1976 when there were extreme deficits across the UK. Periods of regionally low flows occurred in Northern Ireland in the mid-1890s, the early 1920s and the 2010-2012 event in southern England, and 1995-1997 in northern England. The period from 1890 to 1910 (referred to by MCW2007 as the 'Long Drought') was a prolonged period of low flows punctuated by periods without flow deficits – e.g. 1903-1905 where above normal flows were recorded across the country.

In general it appears that more intense low flows occurred in the pre-observation period, whilst the 1980-2015 period included more above normal flow episodes in northern regions (particularly western Scotland) indicated by the white spaces in Figure 3. Across the UK hardly any extreme low flows occurred in the 1980s over the 12-month accumulation period (Figure 3), with the decade generally showing mild drought conditions or above normal flows.

At the shorter three month accumulation period, there was more variability of SSI in both time and space (Figure S1), although there were some similarities to the SSI-12 (Figure 3). The 1920s (also identified in SSI-12) show intense low flow events in southern England. Severe and extreme low flows can be seen in the pre-observation period, and fewer events occurred during the 1980s to early 2000s (except in western Scotland). Although slightly obscured by the very fine-scale variations shown in Figure S1, some spatial coherency emerges for SSI-3 (Figure S1). For example, the 1976 drought, intense in Southern England and Anglian regions, extends northwards and is apparent across the UK. The events of the mid-1930s occur across the UK with the lowest flows in Northern Ireland. The 1995-1996 drought occurs across England and Wales, and to some extent in Scotland – whilst it highlighted longer-term deficits across northern England for SSI-12. At the shorter three month accumulation period, the distinction between Southern England and Anglian regions and the rest of the country is more apparent than at the 12 month accumulation period, with more space-time consistency in SSI values in the south east of England.







**Figure 3 Heatmap of SSI-12 for LFBN catchments (arranged broadly north to south on the y-axis) from 1891 to 2015. Regions are marked for clarity.**



### 3.2 Characterising the severity of past events

The extracted drought events and their characteristics for the nine case study catchments are shown in
Figure 4 for SSI-12 and Figure S2 for SSI-3. For both accumulation periods, events tend to be longer and
less frequent in more southerly catchments regardless of the SSI accumulation period. Events with a lower

5   (i.e. more severe) maximum intensity tend to occur in the pre-observation period, a similar pattern can be
seen for mean deficit, particularly in more northerly catchments. Maximum intensity and mean deficit
seem unaffected by duration, with severe events occurring over the range of durations plotted. On the
Cree, Allan Water and Lud events were clustered in time (e.g. in the 1930s-1950s, before the 1977 and
after 1962, respectively), elsewhere events were more evenly spaced through time. Shorter events tend to

10  have occurred in the last 30 years on the Crumlin and in Scotland, whilst SSI-3 events were longer after
the 1970s on the Lambourn.

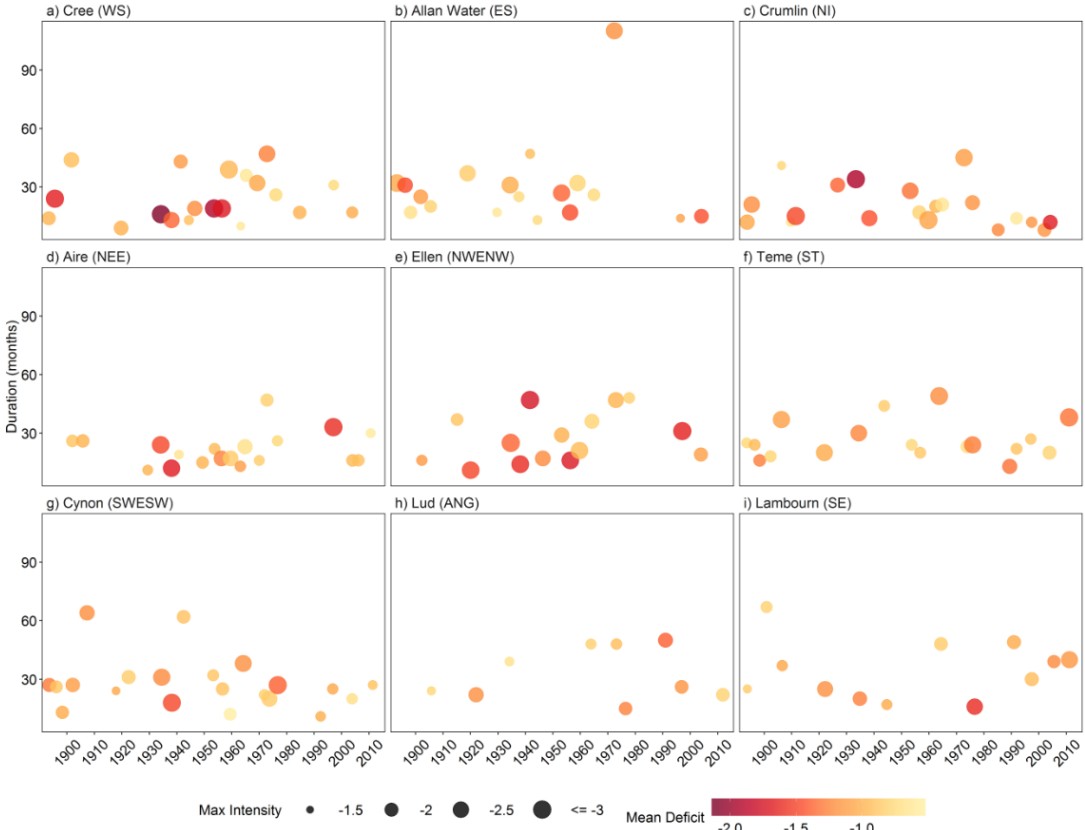

**Figure 4 Extracted events from SSI-12 and their characteristics for the nine case study catchments, plotted
at the midpoint of the event. The size of each point is proportional to the maximum intensity and the colour**
15  **indicates the mean deficit.**



### 3.3 Ranking historic drought events

The longest SSI-12 events varied by location (Figure 5a), but the longest events mostly occurred before 1990 and were clustered between 1940 and 1980.The SSI-12 events of the early to mid-1970s were, for the most part, the longest in northern England, north Wales, Northern Ireland and Scotland ( Figure 5a).

Events in the 1960s ranked highly across the UK (with the exception of Scotland), the 1970s in the north of the UK and the 1900s in the south of England. Many of the 10 longest events occurred during the 'Long Drought' period from the 1890s to 1910s but in many cases these were not the top ranking (i.e. most severe) events.

The event rankings for accumulated deficit were similar to those for duration ( Figure 5a and  Figure 5b)
as longer events are likely to have greater accumulated deficit. When the accumulated deficit is divided by the duration to produce the mean deficit however, a different picture emerges ( Figure 5c). The 1975-1976 event stands out as being highly ranked in terms of mean deficit in southern England and Wales. Events in the mid-1930s rank in the top three across the country. Other severe drought events occurred in the 1950s across northern Britain, in the late 1990s in northern England and Wales as well as the mid-
2000s in some catchments in eastern Scotland, whilst the rank of events in the 1900s and 1960s decreases when looking at mean deficit compared accumulated severity. Several events occurring during the 'Long Drought' period (1890-1910) ranked in the top 10 when considering mean deficit, especially in southern England. The 1920s in south-east England and the 1930s, nationally, stand out dramatically when events are ranked by the maximum intensity (i.e. the lowest monthly SSI value in the event; Figure 5d). This
drought characteristic, more than the others shows a propensity to more severe events in the earlier record than other characteristics.





**Figure 5 Top 10 extracted events from SSI-12 using a threshold of -1.5 for each drought event characteristic. Catchments are arranged roughly north to south on the y-axis with each row representing a catchment; bars represent the top 10 events and are coloured according to the event rank. Darker shades represent higher ranking (i.e. more severe) events.**





For SSI-3, more recent events such as those in the 1990s, 2004-2006 and 2010-2012 rank highly in Anglian and South-East England regions. The drought of 1975-1976 is ranked highly in terms of duration in northern and western regions, but at this shorter accumulation period ranks lower in south-east regions (Figure S3a). When accumulated deficit is considered (Figure S3b), the 1920s ranks in the top half of the rankings across regions of southern England and Wales, whilst the 1930s is coherently ranked in the top 10 across the country (and was particularly highly ranked in Northern Ireland and the south of the UK). The drought of 1995-1997 is highly ranked in the regions of northern England and East Anglia, with the latter being particularly severe for 1990s as well. When ranked by mean deficit, events such as the early 1920s rank highly in south-eastern regions, and the 1929 drought ranks highly (and was ranked top in some catchments in eastern Scotland and north-east England; Figure S3c). In contrast to the duration rankings, catchments in South-East England and Anglian regions ranked highly (and top in many) for 1975-1976, whilst at this shorter accumulation period, the summer drought of 1984 appears in the top half of the rankings, particularly in western regions. The late 1920s (1929) also ranks highly in more northerly and westerly regions for the maximum intensity while the early 1920s ranks particularly highly in Anglian and South-East England regions as does 1975-1976 (Figure S3d).

Figure 6 shows the LFBN catchments where the top ranking SSI-12 events for each event characteristic correspond to the major drought events listed in Table 1. Across England and Wales, the long drought (1890-1910) was the longest event with the largest accumulated deficits events, but the most severe event according to mean deficit and maximum intensity in the north of the UK. In contrast, the 1975-1976 event was worse in terms of mean deficit in southern England and Wales, and amongst the longest with the largest accumulated deficit in northern regions. The events of 1920-1922 and 1933-1934 were amongst the worse in terms of maximum intensity in the south-east and west of the UK, respectively. In the north-east coast, the 1990-1992 was overall the worse drought, whilst it was 1995-1996 in central northern England. The 2010-2012 event had the highest maximum intensity in the Welsh borders and some groundwater dominated catchments in the south-east of England. The 1959 and 2004 events were generally not marked as the highest ranking events for any of the characteristics, except in a handful of catchments – at most five individual catchments during the 1959 event for a range of characteristics, and for only three catchments for the 2004 event.





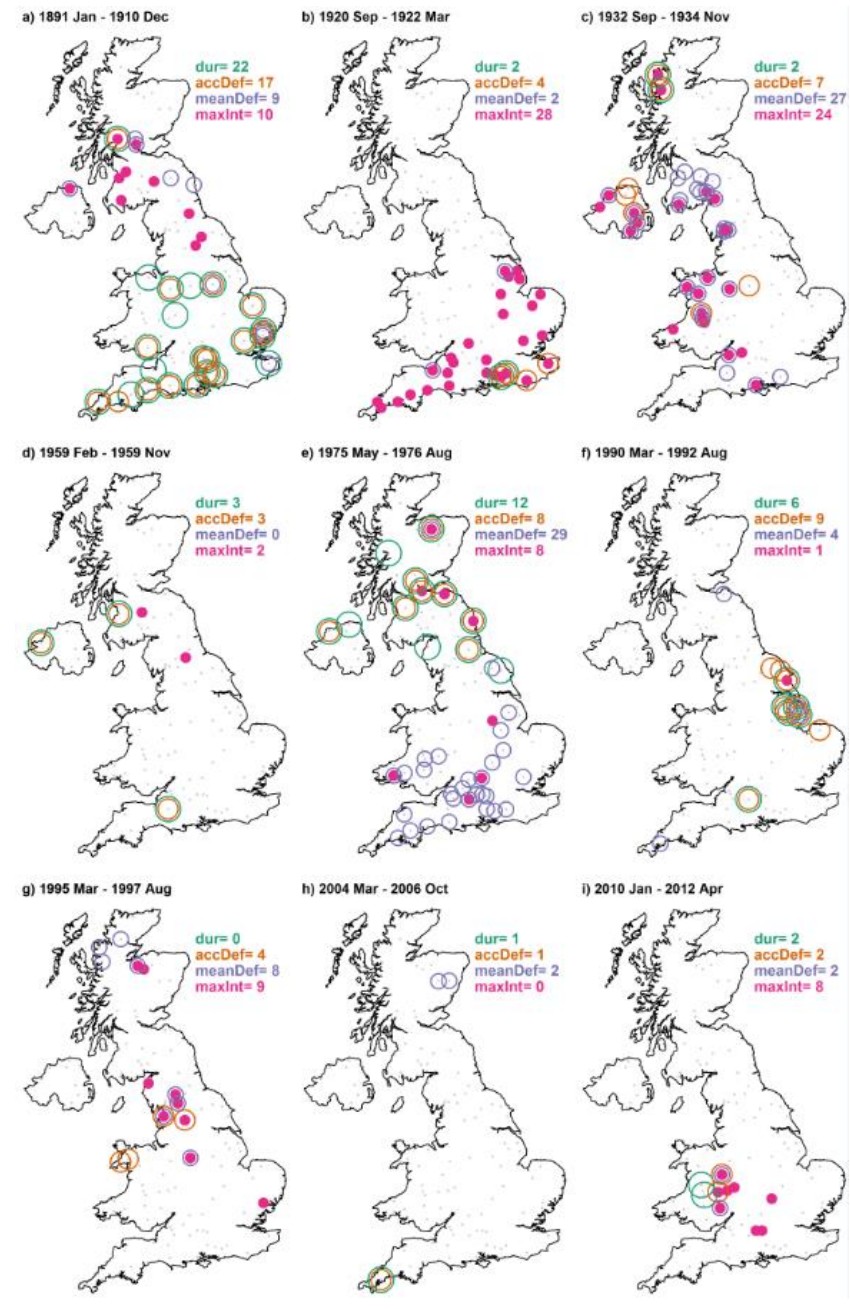

**Figure 6 Location and number of LFBN catchments where the top ranking SSI-12 event corresponds to major events (Table 1) for duration, accumulated deficit, mean deficit and maximum intensity. Each of the nine maps represents one of the major drought events listed in Table 1. Each point on the maps represents the location of the 108 LFBN catchments. Points are coloured pink where the particular event was ranked most severe according to maximum intensity for that catchment. Similarly, points are circled in purple, orange and turquoise to indicate catchments where the particular event was ranked most severe in terms of mean deficit, accumulated deficit and duration, respectively. The numbers in the top right of each map show the number of catchments ranked as most severe for each characteristic for that particular event.**



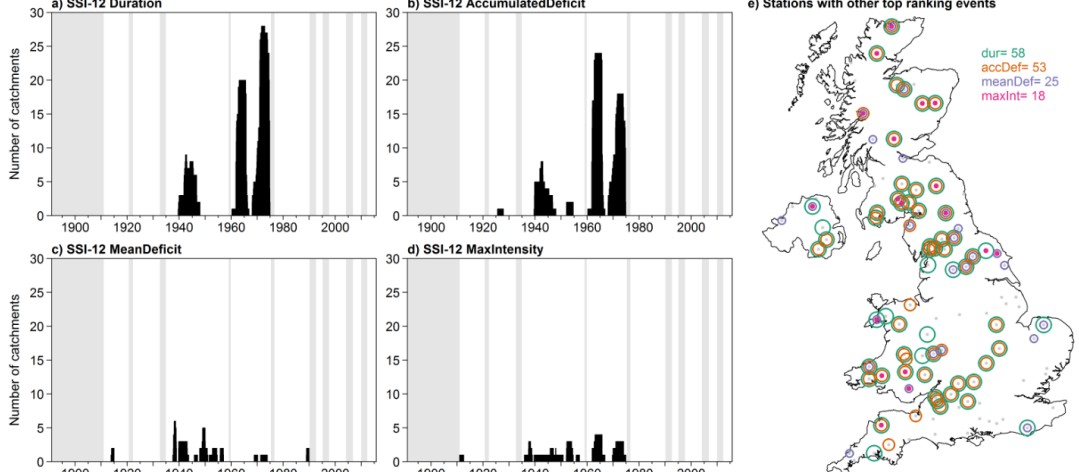

**Figure 7 Months when SSI-12 top ranked events occurred outside of the major events (shaded in grey) for the LFBN catchments and each event characteristic (a-d), and e) the location and number of catchments with other top ranking events for each event characteristic. Points are coloured as described in the caption for Figure 6.**

Figure 7a-d shows the months when the top ranking SSI-12 event did not correspond to the major events for each event characteristic.  The known major drought events exclude top ranking events in the 1940s, 1960s and early 1970s (before the 1975-1976 event) across the four drought characteristics. Figure 7e shows the location of the catchments where the top ranking events occurred outside of the major events, although they occur across the UK, most of these missed events occurred in catchments outside of the south and east of England. A similar spatial pattern can be seen for the top ranking SSI-3 other events (Figure S5e), with a focus in northern and western areas. In contrast to SSI-12, more of the SSI-3 duration and accumulated deficit top ranking events were captured than mean deficit and maximum intensity (around half of which were not captured by the major events). The events not captured by the major for SSI-3 occurred in similar periods as for SSI-12 (i.e. the 1940s, 1960s and early 1970s), with the addition of the late 1920s, late 1930s and 1984.

For both SSI-3 and SSI-12, the period 1980-2015 appears to be well captured by the major events of Table 1 (see Figure S5 and Figure 7). Figure 8 shows three 'other' drought event periods for SSI-12 identified in Figure 7, with top ranking events spread over Great Britain for the 1940s, 1960s and early 1970s. In the 1960s (1960-1966), events were of a longer duration and higher accumulated deficit whilst there were only 4 catchments with top ranking events for maximum intensity and none for mean deficit. Catchments with top ranking events occurring in the 1968-1975 period were focussed in northern parts of the UK with top ranking duration accumulated deficit events spread across Scotland, Northern Ireland and northern England.

For SSI-3, the 1968-1975 period was important for the most severe events according to duration and accumulated deficit in Scotland (Figure S6). Some catchments had top ranking events for more than one



characteristic in this period in Scotland, with just four catchments registering top ranking events (for duration and mean deficit) in England and Wales. Other events were ranked most severe in some catchments and event characteristics (Figure S6); 1928-1929 ranked first for mean deficit and maximum intensity across Scotland and northern England. The drought of the early 1950s ranks first across all four

event characteristics in Northern Ireland, whilst the 1984 event ranked top for mean deficit in 16 catchments in the west of Great Britain.

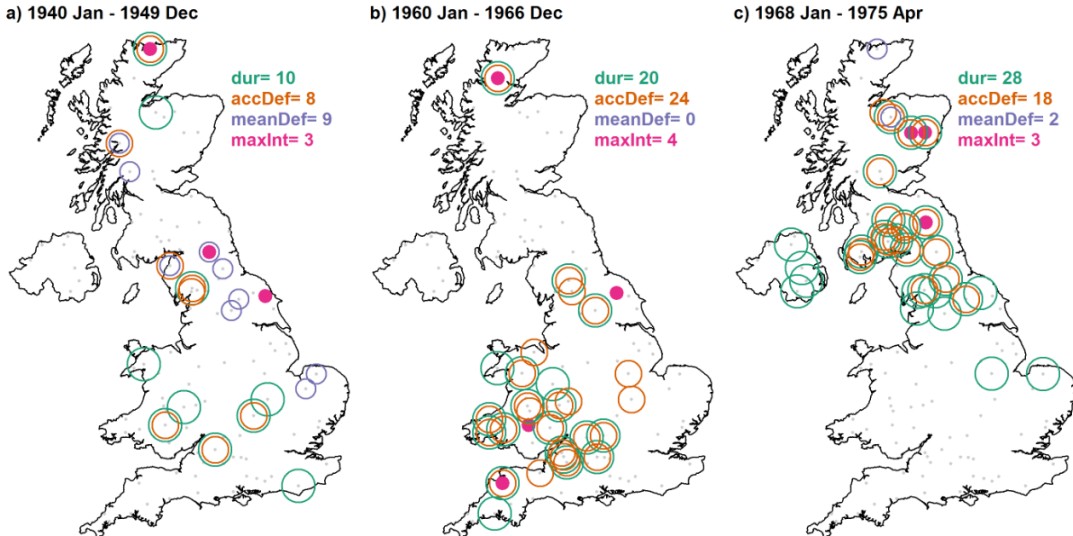

**Figure 8 Location and number of LFBN catchments where the top ranking SSI-12 events for each event characteristic occur in periods outside of the major drought events: 1940-1949, 1960-1966**

**and 1968-1975. Points are coloured as described in the caption for Figure 6.**

Finally we consider the rank of all extracted events for SSI-12 for the major events and 'other' identified events for each event characteristic (Figure 9). By assessing the rank of all the identified events corresponding to the major events (Table 1) and other identified events (Figure 8), the relative severity of the events can be compared. By placing the top 10 ranks in context, we can see that for some events, the

majority of the extracted events fell within the top 10, such as 1995 for duration; 1933 and 1975 for accumulated deficit and mean deficit; and 1920, 1959 and 2010 for maximum intensity. This implies that events such as these were consistently more severe than events with a wider range of ranks or have generally have lower ranks such as1980-1910 or the 1940s. The median rank of 2004 was outside of the top 10 events across all characteristics, as was 1959 for duration and accumulated deficit, and 1891 for

mean deficit and maximum intensity, suggesting that although in some catchments these events were most severe, they were not regualrly ranked highly and so were less severe at the national scale. Most of the major and other events identified from the SSI-3 rank outside of the top 10 (Figure S7), with the exceptions of 1933 for accumulated deficit where the 25-75$^{th}$ percentile of events fall within the top 10. This may be a result of the higher number of shorter events extracted from the SSI-3 series. In some cases,

the median rank of events falls within the top 10, such as 1933 and 1975 for duration, 1975 for





accumulated deficit and 1920 and 1933 for maximum intensity, suggetsing these events were more important a the seasonal scale (SSI-3).

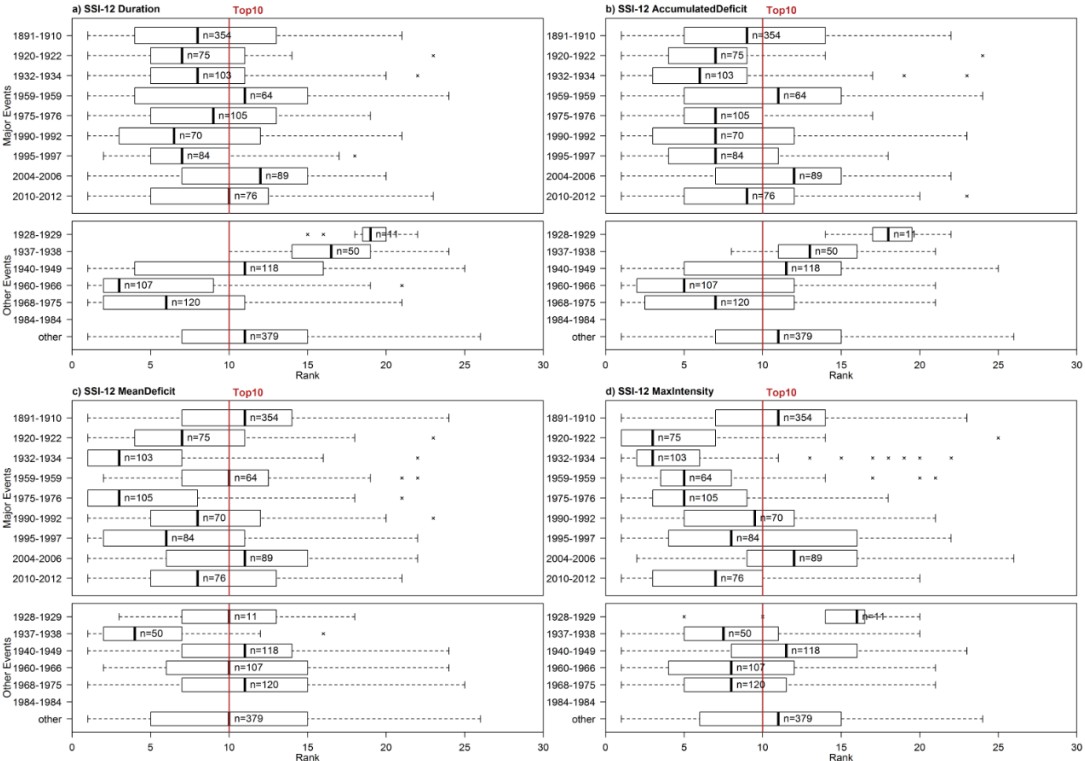

**Figure 9 Boxplots showing the ranks of all SSI-12 extracted events where they overlap with the major drought events (top panel for each event characteristic) and identified 'other' events (bottom panel for each event characteristic). Within each box, *n* refers to the total number of events (across the LFBN) identified that occurred within this period. As multiple events can occur within each given period, it is possible for the value of *n* to be greater than the number of catchments (i.e. 108).**

### 3.4 Evolution and characteristics of major pre-1961 events

While previous work and the above analysis has identified the importance of events in the pre-observation period, their hydrological characteristics have not been fully described at the national scale. The flow reconstructions and derived SSI used here allow a more detailed view of the space-time dynamics of these events comparable with those available for events in the gauged era. Figure 10 shows the SSI for the four earliest events identified in this study prior to 1961: the 'Long Drought' (1891-1910); 1921-1922; 1933-1934; and the 1940s. These events are discussed in more detail in the section below in terms of both SSI-12 and SSI-3; where results pertain to one accumulation period, it has been specified, and otherwise results relate to both accumulation periods.

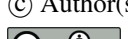



### The Long Drought (1890-1910)

The 20 years period 1890-1910 (the 'Long Drought') showed periods of low flows across much of the country. For SSI-12, there was often spatial coherency in conditions across southern England and Wales, reducing further north (Figure 10), whilst for SSI-3 (Figure S8), only certain periods show national scale

coherency in conditions (such as early 1892, autumn 1892 and 1903-1905). In general however, extreme and severe flow deficits did not occur simultaneously across all regions, e.g. 1895 saw extreme flow deficits were seen across Scotland and Northern Ireland, mild drought in northern England and higher than average flows in the rest of England. With the exception of 1903-1905, northern England was impacted by extreme deficits, whilst several periods of extreme flow deficits occurred in the rest of

England throughout this period. More episodes of severe and extreme deficits can be seen at the seasonal scale using SSI-3 throughout the Long Drought than for SSI-12.

### 1921-1922

The drought of the 1920s was mostly focussed in England and south Wales with severe flow deficits beginning in summer 1921 across southern England for SSI-12 (Figure 10). However, for SSI-3 1920

ended with severe-extreme flow deficits in WS, but the principal period of deficits started across England (with the exception of NWENW) and Northern Ireland in spring 1921. The event continued for a single season in Northern Ireland and North East England, and until winter 1922 in southern England and Wales (Figure S8). The most extreme flow deficits were experienced over the autumn and winter of 1921, and in some catchments in SE extend well into 1922 for SSI-12. In NEE flow deficits were extreme in cases,

with severe deficits experienced throughout 1921. North-western areas again experienced extreme/severe deficits in winter 1923/1924 in Scotland, Northern Ireland, and SWESW for SSI-3.

### 1933-1934

Severe and extreme drought began in winter 1933 in Scotland and Northern Ireland, with much of the country in extreme drought in 1934 (Figure 10 and Figure S8). The east of Scotland, northern England

and northern parts of Anglian region appear to be less affected, although still show at least mild flow deficits. The most severe deficits across the country occur for the duration of 1934, and in some southerly catchments extend into the start of 1935. For SSI-3, deficits ended in the majority of catchments in spring 1934, but continued until the autumn in the south-east of England.

### 1940s

The 1940s was a decade with multiple periods of drought across the country. The decade began with extreme/severe deficits in WS and parts of NEE. Drought conditions were generally mild in other regions with the exception of ANG and SE for SSI-12, where flows were mostly normal or above (Figure 10). During the remainder of the 1940s however, drought was more coherent across the UK in terms of





occurrence (although not in terms of severity) for both accumulation periods. Other notable drought phases in the decade occurred in southern and central England and south Wales in 1944, which extend right into Scotland for SSI-3 (Figure S8); in western Scotland and catchments across northern England, Northern Ireland and north Wales in 1946; across the UK in winter 1947 for SSI-3, and prolonged drought

conditions (albeit mild) across much of England and Wales 1948-1950 with severe drought in NEE in 1949 for SSI-12.

## 4 Discussion

While past studies have identified historic drought episodes in the UK (as summarised in the introduction), a detailed, quantitative assessment of hydrological droughts at the national scale has been

lacking. This paper provides the first systematic characterisation and ranking of hydrological droughts for a period of ~125 years for the UK, using a network of minimally disturbed catchments. In the following discussion, we compare the findings with previous studies, address the scientific and practical significance of the outcomes, before outlining key limitations and recommendations for future research.

### 4.1 Historic hydrological droughts

Understanding historic drought occurrence, duration and severity is vital for drought risk estimation and management in any location, and provides a baseline against which future change can be assessed. For the UK, the primary national scale assessment of past droughts is MCW2007 and the companion report Cole and Marsh (2005). Here, we set our findings in the context of these previous assessments. Unsurprisingly, there is good agreement as to what constitute the most significant events at the national

scale, for example: the Long Drought (1891-1910), 1933-1934, 1975-1976, 1995-1997. However, there are also some important differences. MCW2007 deliberately highlighted national scale events which had evidence of demonstrable societal impact, and so excluded some of the droughts identified here which were either more regionally focussed or lacked the supporting documentary evidence of impacts. Critical droughts for individual catchments may not be those that occurred nationally, so it is important to consider

the most severe droughts on a catchment, or regional, basis. The focus here on characterisation of the identified events for catchments across the UK provides more detail than is provided by MWC2007 who quantified severity for only a handful of long-term hydrometeorological series in the north-west of England and East Anglia.

A the national scale, Jones and Lister (1998) use the drought deficit index to identify droughts in 15

catchments across England and Wales using reconstructed river flows from 1865 to 1993, assessing the severity of the 1989-1992 drought in the context of previous events. Over an 18 month accumulation period of the drought deficit index, the following events ranked as most severe 1975-1976, 1887-1888, 1905-1906, 1921-1922, 1933-1934 and 1943-1944. These events compare well with those identified here,





with the exception of 1887-1888 which is outside of the reconstructed period. Inter-decadal variability is apparent in both sets of reconstructed droughts, with drought rich periods in the 1890s and 1940s. With just 15 catchments, Jones and Lister (1998) could not capture regional- and national-scale events. Here however, the national picture is more developed with space-time evolution of events, and systematic

5   rankings shown for 108 UK catchments for the 125 year period 1891-2015, encompassing the most recent events, with analysis based on reconstructed flows modelled using robust methods (Smith et al., 2019).

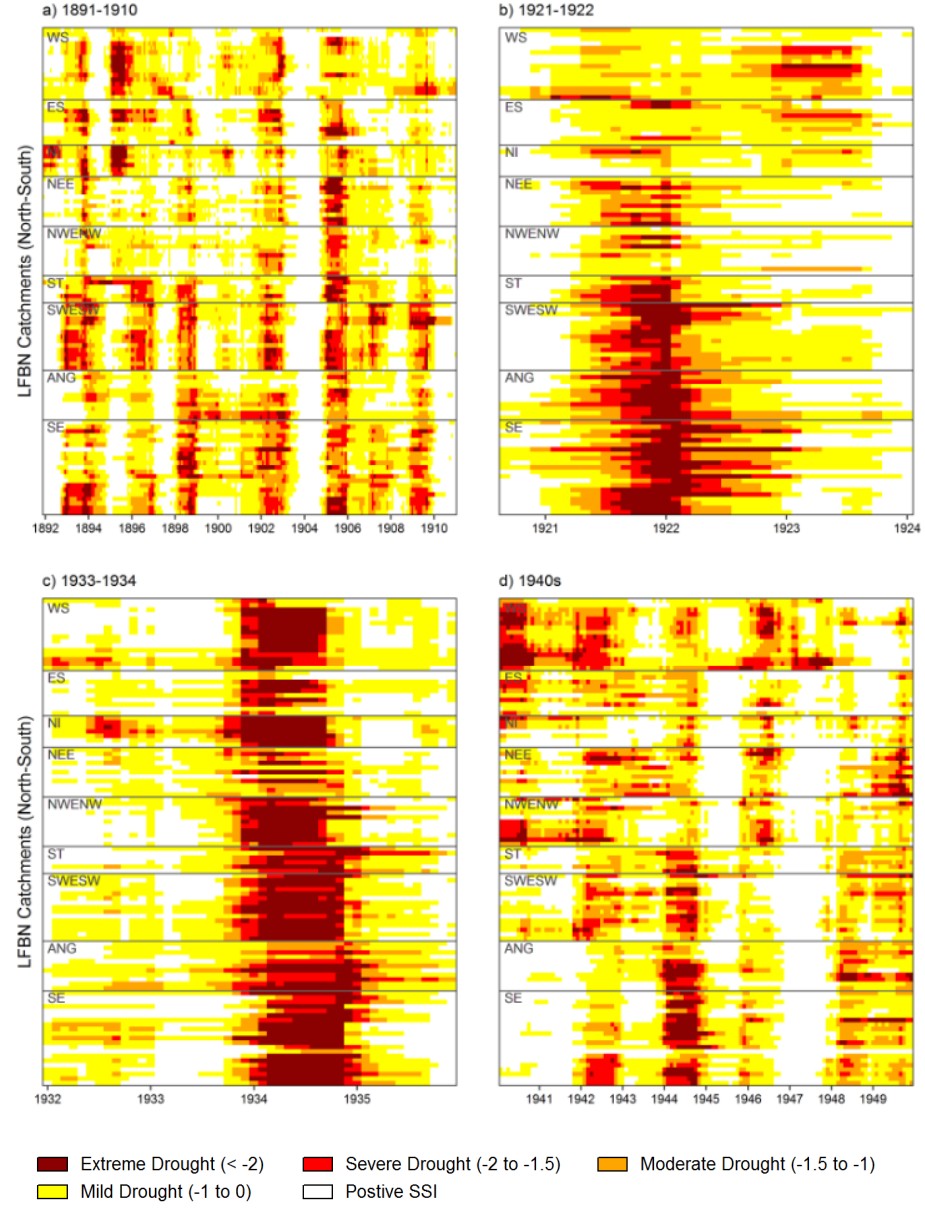

**Figure 10 Heat maps of reconstructed SSI-12 for low flow benchmark catchments, arranged roughly north to south with regions marked for clarity for a) the 'Long Drought' (1890s-1910s), b) 1921-1922, c) 1933-1934 and d) the 1940s.**



Our results also resonate with other historical drought studies in the UK (e.g. Jones and Lister, 1998; Fowler and Kilsby, 2002; Lennard et al., 2016; Spraggs et al., 2015; Rudd et al., 2017). These studies typically focussed on regional assessments using a small number of catchments or gauges. Although there

were parallels with the results shown here (e.g. Spraggs et al. (2015) also find the 1989-1992 event to be the most severe in ANG in river flow reconstructions from 1798-2010; and events identified in Yorkshire by Fowler and Kilsby (2002) corresponded to those in the top 10 rankings for NEE, such as SSI-12 for duration, accumulated deficit and mean deficit: 1905, 1940s, the mid 1960s and 1970s and early 1990s), their transferability is limited by their regional scope and differing methods of assessment.

Parallels with studies in Europe and at the continental scale are also evident, although few studies focus on hydrological droughts at the catchment scale, with most using meteorological drought indicators. Spinoni et al. (2015) used the Standardised Precipitation Index (SPI) to identify and rank events for Europe 1950-2010 and found that for Great Britain more severe events occurred in the earlier part of this time frame, with 1975-1976 ranking as the most severe for Great Britain, whilst 1959 was most extensive.

Van der Schrier et al. (2006) extend further back in time (1901-2002), using meteorological drought indicators based on the Palmer Drought Severity Index. They found that the driest year occurred in 1947 followed by 1921 and 1950, with exceptionally dry summers in England in 1976 and 1921, and in Scotland in 1949, 1945 and 1946. Across the island of Ireland, Noone et al. (2017) found that the most severe events extracted from SPI derived from recovered rainfall data occurred in the mid-to-late 1800s,

but also noted that all of the events within the top 10 rankings for the four event characteristics as used here, occurred before 1977. Although there were fewer top ranking events after the 1975-1976 event here, events of the 1990s and 2010-2012 did rank in the top 10 for both SSI-3 and SSI-12 (Figure S4 and Figure 6, respectively). The lower numbers of drought events post-1980 here and in Noone et al. (2017) are commensurate with increasing trends in runoff in northern and western parts of the UK (e.g. Hannaford,

2015). In terms of hydrological droughts, Sheffield et al. (2009) identified events at the continental scale from global VIC model outputs over 1950-2000: events in the 1960s and 1975-1976 ranked as the most severe across Europe (although were focussed in western and eastern Europe), with much of Europe also affected by drought in 1953-1954. In a low flows assessment using modelled reconstructions for over 650 catchments across France for the period 1800-2012, Caillouet et al. (2017) found that 1976 was the longest

and most severe event in northern France. Given the spatial footprint of the event (Briffa et al., 2009) it is unsurprising that 1976 is similarly highly ranked in southern England in terms of accumulated and mean deficit (e.g. Figure 5 and Figure 6).

It is instructive to consider why the events identified in Figure 8 and Figure S6 were not considered major events by MCW2007 and others. The events of the late 1960s and early 1970s were somewhat

overshadowed by the 1975-1976 event in which there were dramatic impacts on water supplies and the




environment across the UK (e.g. Doornkamp et al., 1980; Rodda and Marsh, 2011). Although the 1975-1976 event was a distinct event, the 20 year period between 1960 and 1979 can be seen to be drought rich in Figure 3 and Figure S1, within which events rank as most severe and fall within the top 10 (Figures 5 and 6, and Figures S3 and S6). In this study the 1940s ranked highly across the different event

characteristics, and affected much of the country with flow deficits occurring somewhere in all months throughout this period (Figure 10 & Figure S1). Although 1943-1944 was classed as a major hydrological event by MCW2007 the documentary evidence to support the physical manifestation of the drought was lacking in the post-War period. As such, the importance of the hydrological droughts of the 1940s is probably understated, as was found by Rudd et al. (2017), and the findings of this study indicate it was a

national scale event (Figure 10d) which may have had substantial impacts on society and the environment.

It is of course important to re-emphasise the hydrological focus of this study. Although the 1959 and 2004 events were identified here, they were not often ranked as most severe (with the exception of 1959 for SSI-12 maximum intensity, Figure 9). These events may be better characterised by rainfall (1959) or groundwater (2004-2006) drought indicators. The addition of impact information (as in MWC2007 or

Noone et al. (2017), for example) would shed more light on these events, the severity of deficits and their impacts (see Section 4.3).

The benefit of using of the two accumulation periods is highlighted when considering the 1984 event, which was not identified using SSI-12, but was ranked as most severe for 16 catchments across western Britain when using SSI-3. Figure 5 and S5 also show the benefit of utilising the different event

characteristics as different events are ranked as most severe when each of the characteristics is considered. This is particularly important for water managers who may be dependent on water sources with differing levels of responsiveness for their supply; such as single season reservoirs, or those with more memory that respond more slowly, such as groundwater dominated river flows.

### 4.2 Applications

The extracted events and new analyses presented here can support further work on the trends and variability of hydrological droughts in the UK. Although work has been undertaken to understand the link between droughts in the UK and large-scale atmospheric forcings (e.g. Folland et al., 2015), this longer, wider set of drought event reconstructions provides a much broader dataset to assess large-scale patterns which cause the clustering of drought events. A better understanding of the relationship between large-

scale atmospheric forcings and drought event characteristics would be useful in the context of drought monitoring, early warning and forecasting applications.

Using reconstructed river flow data (opposed to using observed data) not only extend record lengths on average by ~75 years, but when considering the identified and characterised events provides a much larger



pool of events to work with. Across all catchments, ~67% and ~65% of events extracted from the SSI-12 and SSI-3, respectively) occurred before the observed records began. This highlights the benefit of the long-term view and increasing the pool of events to improve our understanding of past hydrological drought behaviours. Our findings have important implications for those considering hydrological drought

risk, particularly for water resources planners. These reconstructed drought series can be applied in the stress testing of water resource systems for water resources management and drought plans. Most directly, the results can assist in conventional stress tests using the worst droughts on record (e.g. Environment Agency, 2015), but can also inform 'scenario' based stress tests based on synthetic design droughts (e.g. Watts et al., 2012; Anderton et al., 2015). Similarly, the results can provide inputs to – or corroboration

for – the stochastic drought generation techniques that are increasingly used in UK water resource and drought planning (e.g. Atkins, 2016).

Some studies (e.g. Spraggs et al., 2015; Lennard et al., 2016) have demonstrated that extending the hydrological records does not improve water resources planning approaches in particular water supply regions of the UK. However, their regional focus and infrastructure modelling limits their applicability to

other locations where earlier droughts may have substantial impacts on supplies. The results presented here demonstrate that in many regions of the UK, some of the most severe hydrological droughts occurred in the late 19$^{th}$ and early 20$^{th}$ centuries. Further work is required to run such sequences through water supply system models to understand the impacts on drought risk assessments and thus on management plans. But the data and knowledge developed here provides a consistent, national resource for such

studies, which is particularly important as more joined up regional- and national planning is becoming a key priority in the UK (Water UK, 2016; National Infrastructure Commission, 2018). The regional differences in the most severe events over the past ~125 years and the range of event characteristics (i.e. accumulated deficit, duration, mean deficit and maximum intensity) shown here provide a valuable toolkit for assessing hydrological droughts across the country. To this end, results for individual catchments (the

full set of 303 catchments for which reconstructions are available) can also be explored using the 'UK Hydrological Drought Explorer' (https://shiny-apps.ceh.ac.uk/hydro_drought_explorer/), including SSI timeseries, extracted events and the most severe droughts (ranked by the four event characteristics) – see the Data Availability section.

### 4.3 Data limitations & future work

The SSI was derived from daily river flow reconstructions (Smith et al., 2018), extending the gauged record of the LFBN catchments by, on average, more than 75 years, and at most 86 years. Smith et al. (2019) assessed the performance of the modelled flow reconstructions and the derived SSI for 303 UK catchments (including the LFBN). Although the SSI was found to exacerbate any model errors in the flow simulations and the exact magnitude of flow deficits may not be well captured by the reconstructed SSI,

the peaks and troughs and the drought events extracted from the reconstructions compared well to those





from observed flows (using the same identification methodology as used here; Smith et al., 2019). The relative rankings of the extracted drought events here should, therefore, be well captured.

The flow reconstructions of Smith et al. (2018) provide the top 500 ensemble members for each of the LFBN catchments (within the full set of 303 catchments), however, in this study, the single best performing model run was used for each catchment without accounting for model uncertainty. Due to the identified uncertainties in deficit magnitudes in some catchments by Smith et al. (2019), utilising the ensemble data in future studies will provide more confidence in the extraction of drought events that are near the threshold of "severe" drought (i.e. an SSI value of -1.5).

Here, the SSI was calculated using a reference period of 1961-2010 for consistency with companion datasets of gridded Standardised Precipitation Index for the UK (Tanguy et al., 2015; Tanguy et al., 2017). Although this period encompasses well defined flood/drought rich and flood/drought poor periods, the derived SSI and extracted and ranked drought events are derived relative to this period. As high/low rainfall and river flows become more extreme in the future (e.g. Prudhomme et al., 2014), these data should be used with caution for future assessments.

Although reconstruction modelling approaches are valuable, providing otherwise unavailable data for historic events, the limitations of the approach should be recognised. They provide systematic series for the past, but modelling approaches do not address non-stationarities in catchment response or land use change etc. As such, there remains a need for long-term hydrometric data rescue and recovery (e.g. at Wendover Springs, Bayliss et al., 2004; and ongoing efforts to rescue meteorological data e.g. Legg and McCarthy, in preparation), curation (e.g. Dixon et al., 2013), and the incorporation of additional strands of evidence (e.g. documentary, epigraphic and paleohydrological) to supplement and bolster the analysis of hydrological extremes (e.g. Kjeldsen et al., 2014).

The long time series of the SSI and the extracted drought events presented here provide the potential for national scale assessments of trends, changes in timing and seasonality of drought events across the UK. However, it is not only hydrological (streamflow) droughts which are important to understand when managing and planning for drought, meteorological and groundwater droughts also play an important role. Groundwater is critical for UK water supply (particularly in south-east England), and as such it will be necessary to integrate streamflow and groundwater drought reconstruction components to fully assess the impact of the droughts on water supply systems. Recovered and reconstructed data from the Historic Droughts project (Barker et al., 2018b; Bloomfield et al., 2018; Durant and Counsell, 2018) will enable this more holistic analysis of UK drought.

Whilst this paper uses objective, systematic methods to identify, characterise and rank UK hydrological droughts, there was (by design) no consideration of the impacts of each drought. The complex propagation





and development processes of drought events and the effect of management actions may mean that although a drought may be highly ranked in terms of its physical characteristics (duration, accumulated deficit, mean deficit etc.), there may not have been equivalent impacts on the environment or society (e.g. as was found by Lennard et al. (2016) for the Severn Trent region water supplies). However, this paper

provides an independent characterisation of UK droughts which can in future be analysed in conjunction with impact information from a range of sources, for example: the European Drought Impact report Inventory (Stahl et al., 2016); references to drought from legislation (e.g. Lange and Golomoz, 2018), agricultural media (e.g. Rey et al., 2018) or British newspapers (e.g. Baker et al., 2019).

**5 Conclusions**

This study presents timelines of historic reconstructed droughts for 108 near-natural catchments extracted from the Standardised Streamflow Index (SSI) for three and twelve month accumulation periods. It characterises and ranks these past drought events and assesses how relative rankings for each characteristic vary geographically for the first time in the UK. It also provides a fuller understanding of the evolution and characteristics of major, nationally important droughts from the pre-observation period.

The results here reflect the work of previous studies in the UK and at the European scale, identifying well known events as extreme events for the UK (e.g. 1976), but also sheds light on events of the 20th century that have not previously been considered as significant (whether due to a lack of data or evidence of impact), such as the droughts of the 1940s and early 1970s. Results highlight that a range of timescales, or accumulation periods, should be considered when assessing drought severity and hazard in different

locations and for different sectors dependent on water sources with varying response time. By using continuous time series of reconstructed river flow, consistent, objective drought event identification methods and quantitative appraisal of multiple drought characteristics, this study provides a more longitudinal view of drought occurrence and characteristics over a ~125 year period for the UK, with the higher resolution, catchment scale detail important for both science and drought planning applications of

the future.

**Data Availability**

Reconstructed daily streamflow: freely available for download via the Environmental Data Information Centre along with associated metadata on the models performance (Smith et al., 2018). The performance of the model in each catchment, as well as the reconstructed daily river flow timeseries, can be explored

using an interactive web application, the 'UK Reconstructed Flow Data Explorer', at https://shiny-apps.ceh.ac.uk/reconstruction_explorer/.

Standardised Streamflow Index: freely available for download via the Environmental Data Information Centre (Barker et al., 2018b). These SSI data, along with further event analyses can be explored for the



LFBN using an interactive web application, the 'UK Hydrological Drought Explorer', at https://shiny-apps.ceh.ac.uk/hydro_drought_explorer/.

## Author Contributions

Lucy Barker, Jamie Hannaford and Simon Parry discussed and developed the aims of the paper. Lucy Barker was responsible for the data analysis and visualisation. Lucy Barker and Jamie Hannaford prepared the original manuscript, with contributions from Simon Parry, Katie Smith, Maliko Tanguy and Christel Prudhomme.

## Competing interests

The authors declare that they have no conflict of interest.

## Acknowledgements

This research is an outcome of the UK Drought & Water Scarcity Programme Historic Droughts project, financial support was provided by the UK Natural Environment Research Council [NE/L01061X/1].

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
