# Peer review of "Historic hydrological droughts 1891-2015: systematic characterisation for a diverse set of catchments across the UK"

_Hydrology and Earth System Sciences, 2019_

## Referee Comment (RC1) · Anonymous Referee #1 · 26 Jun 2019

The study is in the scope of HESS. I suggest moderate revisions. Figures could be improved (i.e. more clarity, highlight figure message). All together a valuable contribution to the hydrological community!

**Major comments**

A. The paper has a considerable inconsistency in terms of citation style. Please check all the citations to make sure that e.g. Authors et al. (2019), (Authors et al., 2019) and so on is used in a consistent way. This will improve the readability of the paper! Some examples are listed in the technical comments.

B. The reference Legg and McCarthy (prep.) (P05L09) is really problematic for me. As

the readers have no chance to access this paper and "preparation" is for me different to "is submitted", the authors should at least give a short description of what is done in the Legg and McCarthy paper. After all, the model is fed by this data and therefore it is important to understand how meteorological data there is "rescued and digitized". The same is partly true for Smith et al. (2019) as this paper is still under review, isn't it? I suggest to give the reader whenever possible at least a brief description of data/method etc. instead of referring to unpublished studies. I can understand that this is not always easy to do, but it seems to be important to give the reader the chance to understand what has been done. It is also hard to understand how well the model performed (P6L18-L24) in detail, as no further information is given: Here my question is, how valuable is the modelling regarding low flows and streamflow droughts? Here more justification is needed.

C. Regarding the model GR4J I have some concerns regarding the details of the modelling approach. The 4-parameter version is used, if I understand the details in the give references correctly. From Smith et al. (2018) I cannot learn much about the 4 parameters and the functioning, Smith et al. (2019) certainly gives more information on the parameters, but how do you justify that this modelling approach is appropriate for your study propose (i.e. non-stationarity, long series, appropriate for low flows in different seasons)? Especially the slow component and its model representation is of great interest, as the slowest (groundwater) box in the model and its parameterization have potentially a high impact on drought characteristics (such as intensity, duration, deficit). Please comment on this issue (i.e. parameter sensitivity). Are there studies proofing that GR4J is a valuable modeling approach for low flow and drought analysis? Excluding snow and snowmelt processes might be reasonable, but that means that these processes are not relevant for low flows and streamflow droughts in none of the study catchments?

D. A provocative comment: You stated that historical droughts have been more severe than recent droughts (i.e. observed droughts) and a historical assessment is important
to better understand the potential drought magnitude in a region/country. Contrary to that, I would argue that the use of water is adjusted to the water availability of the last, let's say, max. 30-40 years. All water users can only use available water and changes in water availability on a time scale of 3-4 decades influences (of course!) the water uses/water users. So, why is The Long Drought at the beginning of the last century relevant for the water users today? If you show these nice heatmaps with drought severity over 125 years you should also show a heatmap of uncertainty (i.e. comparison between observation period after 1950s and model period before 1950s) (cf. P25L05). Here, I speculate that the uncertainty assessment will soften your statements about historical drought magnitude, duration, intensity.

**Minor comments**

- P02L05-10: How is the statement "historical records are still of fundamental importance in drought planning" justified? From my perspective Brown et al. highlights the lack of historical analysis, but the authors also referred to other studies in paper. However, I suggest to strengthen the study motivation here with more details on the value of historical data or analysis.

- P06L17-20: Would be helpful to give some more information about the criteria used to evaluate the performance.

- P06L26-30: What is the justification to select particularly these nine case study catchments? It is also not clear why case study catchments are used?

- P02L11: Just a suggestion: Are there some reference studies that have investigated major, severe droughts in UK? Could the paragraph better be linked to the P03L15-25) where some historical investigations have been listed?

- P02L20: Is it warm/dry or warm and dry weather?

- P07L04: "end-month"? Is this the same as "right-aligned"?

- Sect 2.2.: I get the idea to have a short- and a long-term analysis (3 and 12 months). However, have you tested other accumulation periods? Is 12 month long enough to capture also long-term anomalies in the slowly reacting, GW-dominated systems in South East England? As events with "less than three months were removed" (is this <3 month or ≤3 month?), I wonder why the SSI-3 is used (as also a "seasonal focus" of the study is stated (P07L29) (see also comment below).

- What means "broadly north to south" exactly (P09L04)? Have you tried the heatmap with squares instead of rectangles (and with a fine border/stroke around the squares; this could improve the clarity of the graph, perhaps.). It would be also interesting to sort the catchments within each geographical group. North-to-south is perhaps not really hydrological meaningful; what about a sorting along a low flow metric (e.g. Q90/Q50) to highlight differences in on-set and termination?

- Fig.4: Are the differences between maximum intensity (dot size) and mean deficit (colors) discussed?

- I am not an expert for historical droughts in UK, but is "The Long Drought" really a 20 year event without drought termination / interruptions? From Fig. 3 and Fig 10a, I have the impression that there are also a lot of "yellow" and "white" segments in the heatmap (e.g. 1904 wasn't really a dry year).

- Fig. 6 is really a nice idea, but it is hard to understand and it take me a while to understand the encodings used in the Figure. I suggest to use a UK-matrix with 9 columns (i.e. events) and 4 rows (i.e. drought characteristics). Then in each subplot all catchments with mild grey dots overplotted by the top ranking catchments in black color. Would improve the clarity of the Fig.

- Would be interesting to quantify the differences between the MCW2007 drought magnitude and the (more severe) droughts on catchment or regional scale (Sect

4.1), e.g. what is the difference of a very critical drought situation in a specific catchment compared to the "national" drought magnitude?

- The authors stated that SSI-3 and SSI-12 are a good choice to identify different drought types (P23). Is this a general recommendation for other studies (3- and 12-months)? If not, what might be a good (and sufficient) set of different SSI-n to capture the variability of historical droughts?

- Sect 4.3 is a little bit long and could be more condensed. The authors discussed potential limitations of their work (e.g. non-stationarity, model uncertainty), but here I missed a clear link to the (own) study results.

**Technical comments**

1. P06L05: Smith et al. (2019) also assessed

2. P06L09: by Smith et al. (2018)

3. P06L11: Low Flow Benchmark Network (LFBN).

4. P06L17: reconstructed by Smith et al. (2018), which include the LFBN, performed

5. For readers from outside UK a short explanation of "Anglian" would be helpful (P09L23).

6. P11L03-04: two times "accumulation period"?

7. lower maximum intensity is more severe? (P11L04/05). Terms should be revised here.

8. Fig.4: The 45 degree axis labels are hard to read, thin grid lines or a lollipop graph instead of bubble graph could improve the readability. If you referred to pre-obs and obs-period than a vertical line to distinguish both periods would be beneficial. Have you tried a lollipop chart here, i.e. vertical lines between dots and x-axis might improve the readability?

9. Remove leading white spaces in (*Figure 5...) on page 12.

---

## Referee Comment (RC2) · Anonymous Referee #2 · 16 Jul 2019

The authors present a study of hydrological drought events over the 1891-2015 period utilizing newly established datasets. I think it is a valuable contribution to the hydrological and low flow and drought community. I would suggest to publish the paper after some minor revisions.

General comments:

Little information is given on the basic datasets used for driving the hydrological model. Please elaborate in more detail on the digitized meteorological data. Is this raw data or have they undergone a homogenization procedure? I also think that a reference for a paper in "preparation" is not suitable. Moreover I think that there has to be a more

in-depth description of the hydrological modelling. E.g. Smith et al. (2019) used six evaluation metrics some of the specific for low flows. What are these metrics and what is the performance? Please provide some information in this respect.

You use the SSI as a standardized hydrological drought indicator. What about the uncertainties considering the fitting of the distribution and how do these translate in terms of derived drought metrics? Since you use mostly rankings of the top events it is rather crucial how the fitting performs particularly at the tails of the distribution. Could you just exemplarily give an indication of possible change in the ranking of some drought metric from fitting uncertainty?

Figure layout: For Figures 3, 5 and 10 I suggest to place the acronyms for the region outside the plot area along the y-axis for better readability. Also rethink the arrangement of catchments along the y-axis, perhaps there is a better way than a strict North/South (driven by climate) alignment (e.g. low flow characteristics). Figure 5: The colorbar as a gradient from red to yellow is in general appropriate for this kind of data in terms of figure layout guidelines. However, since the displayed data is a ranking, I think that the reader would like to see first of all where the top ranked events are. This is not easy in this case. Perhaps you could try a colorbar with more colors? (in R: RColorBrewer palette "Spectral") Or combine two colorbars, one for the top 3 (or 5?) and one for the rest. Don't know how it would look, but it is perhaps worth a try to get the essential information better across.

Several times across the manuscript I stumbled over the terms droughts, drought event or drought periods. I'd like to see more consistency with these terms. The list of major droughts (Table 1) is mostly termed events, however, the 1890-1910 period is not an event from an event definition point of view. This comes rather clear in Figure 10a, where the "long drought" is clearly made up of several individual events(!) all of them with a distinct beginning and end. On the other hand, 1921 (Figure 10b) is clearly an event itself, it has a distinct beginning and end. I suggest to define the names of the major droughts as in Table 1 and stick to the terms, e.g. "1890-1910 drought
period", "1921 event", "1976 event", etc. I think that an event stretching over several years could be termed as the "year xxxx event", with the year being that with maximum drought intensity for example, which has to be defined obviously.

Specific comments:

P2L20: "...short periods of warm and dry weather..."

P2L24: "Moreover, greater climatic variability could mean an increase in persistent blocking episodes and multi-year droughts" please provide a reference for this statement.

P6L5: "Smith et al. (2019) also" please be generally careful with the citations, there are some other inconsistencies.

P6L11: "Low Flow Benchmark Network (LFBN)"

P11L3: suggestion: "For both time scales considered, events tend..."

P19L6: "... e.g. 1895 saw extreme flow deficits across Scotland and Northern Ireland..."

P22L10: In this section some recent research would be appropriate to cite, since there are some events detected in the present paper also listed as extreme droughts in other regions of Europe for example in:

Hanel, M., Rakovec, O., Markonis, Y., Máca, P., Samaniego, L., Kyselá, J., & Kumar, R. (2018). Revisiting the recent European droughts from a long-term perspective. Scientific Reports, 8(1). https://doi.org/10.1038/s41598-018-27464-4

Haslinger, K., & Blöschl, G. (2017). Space-Time Patterns of Meteorological Drought Events in the European Greater Alpine Region Over the Past 210 Years. Water Resources Research, 53(11), 9807–9823. https://doi.org/10.1002/2017WR020797
* * *
202, 2019.

---

## Author Comment (AC1) · 16 Jul 2019

[12pt]article
The study is in the scope of HESS. I suggest moderate revisions. Figures could be improved (i.e. more clarity, highlight figure message). All together a valuable contribution to the hydrological community!

**We would like to thank the reviewer for the positive feedback on our manuscript and are grateful for the comments on how it can be improved. Here, we respond to each comment in turn, the author responses to each point are given in bold below.**

**Major Comments**

A) The paper has a considerable inconsistency in terms of citation style. Please check all the citations to make sure that e.g. Authors et al. (2019), (Authors et al., 2019) and so on is used in a consistent way. This will improve the readability of the paper! Some examples are listed in the technical comments.

**We will ensure that the citation style is correct in the revised version of the paper.**

B) The reference Legg and McCarthy (prep.) (P05L09) is really problematic for me. As the readers have no chance to access this paper and "preparation" is for me different to "is submitted", the authors should at least give a short description of what is done in the Legg and McCarthy paper. After all, the model is fed by this data and therefore it is important to understand how meteorological data there is "rescued and digitized". The same is partly true for Smith et al. (2019) as this paper is still under review,

isn't it? I suggest to give the reader whenever possible at least a brief description of data/method etc. instead of referring to unpublished studies. I can understand that this is not always easy to do, but it seems to be important to give the reader the chance to understand what has been done. It is also hard to understand how well the model performed (P6L18-L24) in detail, as no further information is given: Here my question is, how valuable is the modelling regarding low flows and streamflow droughts? Here more justification is needed.

**We appreciate that we shouldn't cite in preparation/in review papers. The Smith et al., (2019) paper has now been revised and accepted in HESS and the reference will be updated in the revised paper. In regards to Legg McCarthy (in prep) we will remove this 'in preparation' reference and replace it with the references for the finalised data sets (i.e. Met Office 2018, 2019). These datasets have benefitted from additional daily data from ongoing digitisation of daily climatological returns from UK observing stations held in the paper records of the National Meteorological Archives. We feel that citing the catalogued datasets is more appropriate than adding the detail of how data were digitised in this paper. There is a paper in preparation which will sit alongside Met Office (2018) – Hollis et al., however it is still at the 'submission' stage and so it may not be appropriate to cite this paper.**

**In regards to the modelling, the final version of Smith et al. (2019) assesses the performance of the modelling approach for low flows, and the model has been applied in a range of settings, including the UK – see also response to point C below.**

C) Regarding the model GR4J I have some concerns regarding the details of the modelling approach. The 4-parameter version is used, if I understand the details in the give references correctly. From Smith et al. (2018) I cannot learn much about the 4
parameters and the functioning, Smith et al. (2019) certainly gives more information on the parameters, but how do you justify that this modelling approach is appropriate for your study propose (i.e. non-stationarity, long series, appropriate for low flows in different seasons)? Especially the slow component and its model representation is of great interest, as the slowest (groundwater) box in the model and its parameterization have potentially a high impact on drought characteristics (such as intensity, duration, deficit). Please comment on this issue (i.e. parameter sensitivity). Are there studies proofing that GR4J is a valuable modeling approach for low flow and drought analysis? Excluding snow and snowmelt processes might be reasonable, but that means that these processes are not relevant for low flows and streamflow droughts in none of the study catchments?

**We feel the results of Smith et al. (2019) demonstrate that GR4J is appropriate for use at low flows for UK catchments. The multi-objective ensemble modelling approach is covered in detail by Smith et al. (2019) and as such is beyond the scope of this paper, however, we will add some more detail on the use of the model for drought and issues of non-stationarity in long-time series to the revised paper. GR4J has also been used across a wide range of flow regimes around the world, has been used for low flow reconstructions (Caillouet et al., 2017), has demonstrated good performance in a diverse set of catchments in the UK (Harrigan et al., 2018), and good performance at simulating temporal transitions between wet and dry periods (Broderick et al., 2016).**

D) A provocative comment: You stated that historical droughts have been more severe than recent droughts (i.e. observed droughts) and a historical assessment is important to better understand the potential drought magnitude in a region/country. Contrary to that, I would argue that the use of water is adjusted to the water availability of the last, let's say, max. 30-40 years. All water users can only use available water and changes in water availability on a time scale of 3-4 decades influences (of course!)
the water uses/water users. So, why is The Long Drought at the beginning of the last century relevant for the water users today? If you show these nice heatmaps with drought severity over 125 years you should also show a heatmap of uncertainty (i.e. comparison between observation period after 1950s and model period before 1950s) (cf. P25L05). Here, I speculate that the uncertainty assessment will soften your statements about historical drought magnitude, duration, intensity.

**We appreciate your point that drought events may not have the same impacts now as they have done previously due to more resilient water supply and management systems. But regardless of water use, water resource managers look at natural water availability in their drought management plans. In the past UK, water supply drought plans have been based around planning for the worst event on record, and water companies must now plan for events outside of the historic record. Critical to these approaches is an understanding of events that have occurred in the past. Here we have identified past instances of events where natural water availability has been significantly lower, and for longer time periods than we have experienced in the recent past. Despite adjustments in water use to availability, extreme water deficits will still impact society, so information to better inform water resource managers on the characteristics of such events will always be valuable. The additional data provided by the reconstructed flow data provide this long view and enable the consistent identification and characterisation of droughts over the past 125 years.**

**Regarding uncertainty - using all 500 model parameterisations from the Smith et al. (2018) dataset was beyond the scope of this study for computational reasons and we acknowledge this in the paper (P25L3) and will further clarify this in the revised paper. However Smith et al. (2019, Figs 9  10) assessed the uncertainty of extracted drought events in the modelled timeseries for nine case study catchments. They found that overall the model results provided accurate**

simulation of drought events, and that uncertainty was higher in the timing of events, whilst estimates of the accumulated deficit were in better agreement. This reinforces the benefit of these data in characterising drought duration, magnitude and intensity.

**Minor Comments**

- P02L05-10: How is the statement "historical records are still of fundamental importance in drought planning" justified? From my perspective Brown et al. highlights the lack of historical analysis, but the authors also referred to other studies in paper. However, I suggest to strengthen the study motivation here with more details on the value of historical data or analysis.
**This section of the introduction was intended to introduce the benefits of using of historic data in planning approaches; later in the introduction for example on pages 3 and 4, the motivation of this study is more clearly defined.**

- P06L17-20: Would be helpful to give some more information about the criteria used to evaluate the performance.
**The following metrics were used by Smith et al., (2019) to assess model performance:**

    – **NSE – good at magnitude and timing of peak flows**
    – **LogNSE – NSE on log flows in an attempt to match magnitude and timing of lower flows**
    – **MAPE – overall magnitude of variability**
    – **absPBIAS – total water balance**
    – **MAM30APE – error in the lowest of flows**

– **Q95APE – fitting the tail of the FDC**

**These metrics, which although assess performance across the flow regime, have a slight bias towards low flows, but as this is covered in the now revised and accepted Smith et al., (2019) we will continue to direct the reader to this paper.**

- P06L26-30: What is the justification to select particularly these nine case study catchments? It is also not clear why case study catchments are used?
  **The paragraph (and subsequent paragraphs) mentioned on P3 already describe the previous assessments of historical droughts in the UK, here we simply introduce that droughts do occur in the UK, however, we will add an exemplar reference to P2L11 in the revised paper.**

- P02L20: Is it warm/dry or warm and dry weather?
  **This should read 'warm and dry' and will be changed in the revised paper.**

- P07L04: "end-month"? Is this the same as "right-aligned"?
  **The SSI is calculated for the end month of the accumulation period, i.e. SSI-3 in December is the SSI for the period October-December. As we have used this terminology previously (e.g. Barker et al., 2016 and Svensson et al., 2017), we will keep this notation but will provide an example for the 3 and 12 accumulation period in the revised paper.**

- Sect 2.2.: I get the idea to have a short- and a long-term analysis (3 and 12 months). However, have you tested other accumulation periods? Is 12 month long enough to capture also long-term anomalies in the slowly reacting, GW dominated systems in South East England? As events with "less than three months were removed" (is this <3 month or <=3month?), I wonder why the SSI3 is used (as also a "seasonal focus" of the study is stated (P07L29) (see also comment below).

**To clarify, we removed events with a duration of less than three months (i.e. durations of 1 or 2 months). The SSI-3 was taken to be analogous to seasonal deficits as UK seasons are generally determined to be around three months long. SSI-12 was selected as it encompasses deficits over multiple seasons, representing longer term deficits. Drought impacts occur at a range of time scales across the UK (e.g. Bachmair et al. 2016), and we did run the analysis for additional accumulation periods (1, 3, 6, 9, 12, 18 and 24 months) but felt that this was too much information to present in one paper, and so here we selected the 3 and 12 month SSI to broadly represent short and long-term droughts. Results from other accumulation periods can be explored using the UK Hydrological Drought Explorer mentioned on P24L25-26. We will ensure it is clear results for additional accumulation periods can be accessed here too as well as individual catchment results.**

- What means "broadly north to south" exactly (P09L04)? Have you tried the heatmap with squares instead of rectangles (and with a fine border/stroke around the squares; this could improve the clarity of the graph, perhaps.). It would be also interesting to sort the catchments within each geographical group. North-to south is perhaps not really hydrological meaningful; what about a sorting along a low flow metric (e.g. Q90/Q50) to highlight differences in on-set and termination? **Gauging stations in the UK are assigned station IDs by the UK National River Flow Archive based on the Hydrometric Area in which the station and catchment falls (for more information see the National River Flow Archive website here: https://nrfa.ceh.ac.uk/station-number and https://nrfa.ceh.ac.uk/hydrometric-areas). The catchments were therefore sorted by station number so they were ordered by region in the following order: Western Scotland, Eastern Scotland, Northern Ireland, North East England, North West England and North Wales, Severn Trent, South West England and South Wales, Anglian and Southern England. Within these regions the indi-**

[Figure]

vidual catchments may not be strictly ordered north to south, but the catchments are in general, therefore ordered by areas that are climatically and physically similar. We appreciate that the catchments could be arranged in a number of ways but we don't think this would add significantly to the interpretation of the figure.

- Fig.4: Are the differences between maximum intensity (dot size) and mean deficit (colors) discussed?
  **We will ensure that the differences between maximum intensity (point size) and mean deficit (point colour) are discussed in Section 3.2 in the revised paper.**

- I am not an expert for historical droughts in UK, but is "The Long Drought" really a 20 year event without drought termination / interruptions? From Fig. 3 and Fig 10a, I have the impression that there are also a lot of "yellow" and "white" segments in the heatmap (e.g. 1904 wasn't really a dry year).
  **The Long Drought was indeed a period of many shorter deficits and 1903-1905 was a wet interlude in this prolonged dry period. We explain this in both Section 3.1 and Section 3.4, and we introduce that this period (1890-1910) was called the 'Long Drought' by Marsh et al. (2007) on P2L12.**

- Fig. 6 is really a nice idea, but it is hard to understand and it take me a while to understand the encodings used in the Figure. I suggest to use a UK-matrix with 9 columns (i.e. events) and 4 rows (i.e. drought characteristics). Then in each subplot all catchments with mild grey dots overplotted by the top ranking catchments in black color. Would improve the clarity of the Fig.
  **We would prefer not to separate out the event characteristics to separate maps in Figure 6, but we will add a key to indicate which characteristic relates to which point size and colour, we think that this will make the figure easier to interpret. We will also add this key to the subsequent plots of a**

**similar style.**

- Would be interesting to quantify the differences between the MCW2007 drought magnitude and the (more severe) droughts on catchment or regional scale (Sect 4.1), e.g. what is the difference of a very critical drought situation in a specific catchment compared to the "national" drought magnitude?
**The focus of this paper was the drought events identified, characterised and ranked at the catchment scale in a consistent way. MCW2007 did not undertake any systematic quantification of drought magnitude, and so we compared the top ranking events we identified to the major events of MCW2007 e.g. Figure 6. We feel it is beyond the scope of this paper to assess regional-national drought severity/magnitude, and this will be the focus of further works which will assess national/regional drought severity in relation to historic drought impacts.**

- The authors stated that SSI-3 and SSI-12 are a good choice to identify different drought types (P23). Is this a general recommendation for other studies (3- and 12-months)? If not, what might be a good (and sufficient) set of different SSI-n to capture the variability of historical droughts?
**We use the 3 and 12 month accumulation period to characterise single season (3-month) and multi-season annual (SSI-12) hydrological droughts. However, the exact choice of accumulation period in future studies will depend on the motivation and application of research.**

- Sect 4.3 is a little bit long and could be more condensed. The authors discussed potential limitations of their work (e.g. non-stationarity, model uncertainty), but here I missed a clear link to the (own) study results.
**We feel that limitations outlined in Section 4.3 are the key limitations of this study and capture some of the issues related to the modelling approach brought up by the reviewer above. As this section also relates to the next**

**steps for this research we feel it is appropriate length, but will endeavour to condense this section in the revised manuscript.**

**Technical Comments**

1. P06L05: Smith et al. (2019) also assessed
   **We will change this citation style in the revised paper.**

2. P06L09: by Smith et al. (2018)
   **We will change this citation style in the revised paper.**

3. P06L11: Low Flow Benchmark Network (LFBN).
   **We will capitalise Low Flow in the revised paper.**

4. P06L17: reconstructed by Smith et al. (2018), which include the LFBN, performed
   **We will modify this text as suggested in the revised paper.**

5. For readers from outside UK a short explanation of "Anglian" would be helpful (P09L23).
   **Anglian region here refers to the Anglian region (ANG) marked on Figure 1 in the paper and on the heatmaps e.g. Figure 3, will refer to this by the region acronym ANG to avoid confusion here.**

6. P11L03-04: two times "accumulation period"?
   **We will reword this sentence to remove the two mentions of accumulation periods in the revised paper.**

7. 7. lower maximum intensity is more severe? (P11L04/05). Terms should be revised here.
   **Here we mean 'lower' as more negative which in terms of the maximum**

**intensity (i.e. the lowest SSI value within the event) would equal a more severe event. We will clarify this in the revised paper.**

8. Fig.4: The 45 degree axis labels are hard to read, thin grid lines or a lollipop graph instead of bubble graph could improve the readability. If you referred to pre-obs and obs-period than a vertical line to distinguish both periods would be beneficial. Have you tried a lollipop chart here, i.e. vertical lines between dots and x-axis might improve the readability?
**We will modify the figure so the axis labels are at 90o to the x-axis and will assess what the best way is to make this plot easier to interpret, e.g. by adding vertical lines connecting the points to the x-axis and annotation to mark the observed and pre-observed period as suggested.**

9. Remove leading white spaces in (*Figure 5. . .) on page 12.
**The white space on page 12 will be removed when the paper is formatted in HESS style following revisions.**

**References**

Bachmair, S., Svensson, C., Hannaford, J., Barker, L., and Stahl, K.: A quantitative analysis to objectively appraise drought indicators and model drought impacts, Hydrology and Earth System Sciences, 20, 2589-2609, 2016.

Barker, L. J., Hannaford, J., Chiverton, A., and Svensson, C.: From meteorological to hydrological drought using standardised indicators, Hydrology and Earth System Sciences, 20, 2483-2505, 2016.

Broderick, C., Matthews, T., Wilby, R. L., Bastola, S., and Murphy, C.: Transferability of hydrological models and ensemble averaging methods between contrasting climatic periods, Water Resources Research, 52, 8343-8373, doi:10.1002/2016WR018850,

2016.

Caillouet, L., Vidal, J. P., Sauquet, E., Devers, A., and Graff, B.: Ensemble reconstruction of spatio-temporal extreme low-flow events in France since 1871, Hydrol. Earth Syst. Sci., 21, 2923-2951, 10.5194/hess-21-2923-2017, 2017.

Harrigan, S., Prudhomme, C., Parry, S., Smith, K., and Tanguy, M.: Benchmarking ensemble streamflow prediction skill in the UK, Hydrology and Earth System Sciences, 22, 2023-2039, 2018.

Hollis, D., M. McCarthy, M. Kendon, T. Legg, I. Simpson, 2019: HadUK-Grid – a new UK dataset of gridded climate observations, Geoscience Data Journal, submitted.

Met Office: HadUK-Grid gridded and regional average climate observations for the UK, Centre for Environmental Data Analysis, 10.5285/4dc8450d889a491ebb20e724debe2dfb, 2018.

Met Office: MIDAS Open: UK daily rainfall data, v201901, Centre for Environmental Data Analysis, 10.5285/ec54d5e5288a4ebb8c7ad2a1ef6aec42, 2019. Marsh, T., Cole, G., and Wilby, R.: Major droughts in England and Wales, 1800–2006, Weather, 62, 87-93, 10.1002/wea.67, 2007.

Smith, K., Barker, L., Tanguy, M., Parry, S., Harrigan, S., Legg, T., Prudhomme, C., and Hannaford, J.: A Multi-Objective Ensemble Approach to Hydrological Modelling in the UK: An Application to Historic Drought Reconstruction, Hydrology and Earth System Sciences Discussions, https://doi.org/10.5194/hess-2019-3, 2019.
Svensson, C., Hannaford, J., and Prosdocimi, I.: Statistical distributions for monthly aggregations of precipitation and streamflow in drought indicator applications, Water Resources Research, 53, 999-1018, 2017.

---

## Short Comment (SC1) · 21 Jul 2019

An excellent and interesting paper and good to see a national scale analysis of historic drought.

I have some general comments which I hope may be helpful:

In the abstract you could say '108 near natural catchments'. I think it would be useful for water managers to know up-front the type of catchment you are analysing as they deal with a whole range of catchment conditions.

page6 lines9-16 To expand on the previous comment, much of the time water resource

planning has to deal with non-natural flows. For this situation your method would require naturalisation of recorded flow series before model calibration and flow series extension. Alternatively, model calibration with artificial influences included. It's obviously achievable but a longer procedure with more room for introducing error.

page6 line23 Would it be worth appending a list in the supplement of the 108 catchments shown in Figure 1? I found myself trying to work out which Anglian catchments were used and others may want to do the same in other parts of the UK.

Figure 2. The Maximum Intensity appears be dimensioned in units of time. Should it be defined by say a horizontal dotted line from the lowest SSI point (-2) to the Y axis, plus a vertical line (with arrowheads at each end) from the dotted line to SSI zero line? The dimension would then be SSI.

page14 lines7-8 1989-90 was severe in East Anglia, particularly for groundwater. So are you saying '...with the latter being particularly severe for the 1990s as a whole'?

Figure 6. Looking at Anglian region it is clear that use of different characteristics identifies different droughts i.e.1891-1910, 1920-22, 1975-6, 1990-92. Water resource modelling often shows that there is sometimes very little difference between major droughts when it comes to defining system yield for use in the supply-demand balance. If you look at Fig. 10 of our 2015 paper you can see that simulated reservoir drawdown at Grafham was very similar for 4 droughts: 1920-1, 1933-4, 1944-7 and 1975-6. Tweaks to the WR model parameters, e.g. frequency of supply restrictions, have been shown to invoke one or other of these droughts as critical. The point I'm making is that only by simulating a WR system over the whole historic series (behavioural analysis) will the critical drought be found - I think you say this later in the paper!

page20 line11 better to stick with 'near natural catchments'?

page22 lines5-6 we found 1989-92 to be the most severe in the north of the Anglian region, including the Lud catchment, not the whole region (2015 paper Table 2). It

ranked only 9th for Alton in Suffolk. But as you point out: different method, different durations.

page23 lines 6-10 1943-46 drought significant in west of Anglian region (2015 paper abstract and Table 2 for Grafham 24 months period)

page24 lines12-13 Extending the hydrological series back from 1920 to 1800 did not introduce different critical droughts - they all remained post-1920 (2015 paper Table 3). It didn't change the approach or methodology, so could you delete 'approaches' and just say '...planning in particular water...'?

page 24 lines17-21 Totally agree! I noted this in our 2015 paper Conclusions point 6.

page 25 lines15-22 'non-stationarities in catchment response or land use change' may not be an issue for water resource planning. Current or projected future (planned) artificial catchment influences can be added to an extended naturalised series for use in water resource models. Catchment change etc. would of course be important for corroboration with documentary evidence.

The first referee questions why the choice of 3 and 12 month SSI. From the water resources planning and management perspective longer droughts have been a concern, notably during the 2010-12 episode, with the ever increasing impact of global warming. So, although drought structure under a changing climate is conjectural, 24 and 36 month SSI would be interesting!

And a few typos:

page2 line34 'quantify and understand'?

page8 line12 delete the ', they'

page 12 line20 you use the word 'record' when technically the earlier data is not recorded, so perhaps say 'period'? Anywhere else in paper?

page16 line14 'major droughts for'?

page17 line18 1890-1910

page17 line 21 'regularly'

page17 line29 At

page23 line19 Should it be Figure 5 or 7?

Kind regards, Gerry Spraggs

---

## Author Comment (AC2) · 29 Jul 2019

**Author Response**

The authors present a study of hydrological drought events over the 1891-2015 period utilizing newly established datasets. I think it is a valuable contribution to the hydrological and low flow and drought community. I would suggest to publish the paper after some minor revisions.

[Figure]

We would like to thank the reviewer for their positive feedback on our manuscript and are grateful for the comments on how it can be improved. Here, we respond to each comment in turn with the author responses to each point given in bold below.

**General Comments**

Little information is given on the basic datasets used for driving the hydrological model. Please elaborate in more detail on the digitized meteorological data. Is this raw data or have they undergone a homogenization procedure? I also think that a reference for a paper in "preparation" is not suitable. Moreover I think that there has to be a more in-depth description of the hydrological modelling. E.g. Smith et al. (2019) used six evaluation metrics some of the specific for low flows. What are these metrics and what is the performance? Please provide some information in this respect.

**Reviewer 1 also had the same concern and we appreciate that we shouldn't cite in preparation papers. In regards to Legg and McCarthy (in prep), we will remove this 'in preparation' reference and replace it with the references for the finalised data sets (i.e. Met Office 2018, 2019). These datasets have benefitted from additional daily data from ongoing digitisation of daily climatological returns from UK observing stations held in the paper records of the National Meteorological Archives. We feel that citing the catalogued datasets is more appropriate than adding the detail of how data were digitised in this paper. There is a paper in preparation which will sit alongside Met Office (2018) – Hollis et al., however it is still at the 'submission' stage and so it may not be appropriate to cite this paper.**

**As stated in our response to Reviewer 1, we will add more detail on the per-**

formance metrics used in Smith et al. (2019) as well as how well the models performed spatially to the revised paper. However, the results of the multi-objective modelling approach, including the performance is described in detail in Smith et al. (2019) and we feel that it is more appropriate to guide the reader to the detailed Figure 3 in Smith et al. (2019) and the performance data available in Smith et al. (2018), than to include detailed information on model performance in this study.

You use the SSI as a standardized hydrological drought indicator. What about the uncertainties considering the fitting of the distribution and how do these translate in terms of derived drought metrics? Since you use mostly rankings of the top events it is rather crucial how the fitting performs particularly at the tails of the distribution. Could you just exemplarily give an indication of possible change in the ranking of some drought metric from fitting uncertainty?

There are clearly uncertainties in the fitting of a distribution (which ever distribution is selected), however we used the Tweedie distribution because it has been shown by Svensson et al. (2017) to fit best for UK flow data for 121 Benchmark Catchments in the UK (the majority of which overlap with the 108 LFBN catchments used in this study). Tweedie was recommended as the distribution of choice for the UK after rigorous testing by Svensson et al. (2017) of 12 distributions (including those commonly used for standardised drought indicators such as Gamma and GEV) with special attention paid to the tails of the distribution (see Svensson et al. 2017 Section 4.3); the Tweedie has the advantage of being a flexible three parameter distribution that has a lower at zero. We will add the benefits of the Tweedie distribution to the revised paper, and add a brief discussion of the uncertainty to the discussion, recognising Reviewer 1's comments about the length of this section we will sure this addition (and the rest of the section) is concise.

Figure Layout For Figures 3, 5 and 10 I suggest to place the acronyms for the region outside the plot area along the y-axis for better readability. Also rethink the arrangement of catchments along the y-axis, perhaps there is a better way than a strict North/South (driven by climate) alignment (e.g. low flow characteristics).

**In the revised paper we will move the region labels to outside the plot area to improve the readability for the plots mentioned.**

**Currently, the catchments are ordered by their gauging station ID number which are assigned based on the on the Hydrometric Area in which the gauging station is located (for more information see the National River Flow Archive website here: https://nrfa.ceh.ac.uk/station-number and https://nrfa.ceh.ac.uk/hydrometric-areas). The catchments are in general, therefore ordered by areas that are climatically and physically similar. However, the arrangement of catchments was also mentioned by Reviewer 1, and we will investigate whether changing the order of the catchments (e.g. ordered by Q95 within regions) improves the interpretation of the plots.**

Figure 5: The colorbar as a gradient from red to yellow is in general appropriate for this kind of data in terms of figure layout guidelines. However, since the displayed data is a ranking, I think that the reader would like to see first of all where the top ranked events are. This is not easy in this case. Perhaps you could try a colorbar with more colors? (in R: RColorBrewer palette "Spectral") Or combine two colorbars, one for the top 3 (or 5?) and one for the rest. Don't know how it would look, but it is perhaps worth a try to get the essential information better across.

**We appreciate it is not always easy to see which is the top ranking event on these plots. We would prefer not to use the spectral colour palette because as**

**it includes green and red can cause problems of interpretation for those with colour blindness, the colour palette we used also works when printed in grey scale. We will however, work to improve this plot and ensure it is easy to see when the top ranking event(s) occurred in the revised paper, e.g. by adding additional colours to the palette.**

Several times across the manuscript I stumbled over the terms droughts, drought event or drought periods. I0d like to see more consistency with these terms. The list of major droughts (Table 1) is mostly termed events, however, the 1890-1910 period is not an event from an event definition point of view. This comes rather clear in Figure 10a, where the "long drought" is clearly made up of several individual events(!) all of them with a distinct beginning and end. On the other hand, 1921 (Figure 10b) is clearly an event itself, it has a distinct beginning and end. I suggest to define the names of the major droughts as in Table 1 and stick to the terms, e.g. "1890-1910 drought period", "1921 event", "1976 event", etc. I think that an event stretching over several years could be termed as the "year xxxx event", with the year being that with maximum drought intensity for example, which has to be defined obviously.

**The Long Drought was indeed a period of many shorter deficits and 1903-1905 was a wet interlude in this prolonged dry period. We explain this in both Section 3.1 and Section 3.4, and we introduce that this period (1890-1910) was called the 'Long Drought' by Marsh et al. (2007) on P2L12 and classed it as a major 'event'. We would prefer to keep the event (or period) names as they are, e.g. 1975-1976, 1921-1923, 1891-1910 as this most clearly captures when the event occurred, but will ensure that the 1891-1910 period is referred to as a 'period' rather than 'event' in the revised manuscript to avoid this confusion.**

**Specific Comments**
P2L20: ". . .short periods of warm and dry weather. . ."

**We will modify this text as suggested in the revised paper.**

P2L24: "Moreover, greater climatic variability could mean an increase in persistent blocking episodes and multi-year droughts" please provide a reference for this statement.

**We will cite Folland et al. (2015) in this sentence in the revised paper.**

P2L24: P6L5: "Smith et al. (2019) also" please be generally careful with the citations, there are some other inconsistencies.

**We will change this citation style in the revised paper and ensure the rest of the references are in the correct style.**

P2L24: P6L11: "Low Flow Benchmark Network (LFBN)"

**We will capitalise Low Flow in the revised paper.**

P2L24: P6L11: P11L3: suggestion: "For both time scales considered, events tend. . ."

**We will modify this sentence in the revised paper.**

P2L24: P6L11: P19L6: ". . . e.g. 1895 saw extreme flow deficits across Scotland and Northern Ireland. . ."

**Thank you for this correction, we will change this sentence as suggested in the revised paper.**

P2L24: P6L11: P22L10: In this section some recent research would be appropriate to cite, since there are some events detected in the present paper also listed as extreme droughts in other regions of Europe for example in:

Hanel, M., Rakovec, O., Markonis, Y., Máca, P., Samaniego, L., KyselÃÂą, J., Kumar, R. (2018). Revisiting the recent European droughts from a long-term perspective. Scientific Reports, 8(1). https://doi.org/10.1038/s41598-018-27464-4

Haslinger, K., Blöschl, G. (2017). Space-Time Patterns of Meteorological Drought Events in the European Greater Alpine Region Over the Past 210 Years. Water Resources Research, 53(11), 9807–9823. https://doi.org/10.1002/2017WR020797
**Thank you for these suggestions of additional references, we will ensure to include them in the revised paper.**

**References**

Folland, C. K., Hannaford, J., Bloomfield, J. P., Kendon, M., Svensson, C., Marchant, B. P., Prior, J., and Wallace, E.: Multi-annual droughts in the English Lowlands: a review of their characteristics and climate drivers in the winter half-year, Hydrol. Earth Syst. Sci., 19, 2353-2375, https://doi.org/10.5194/hess-19-2353-2015, 2015.

Hollis, D., M. McCarthy, M. Kendon, T. Legg, I. Simpson, 2019: HadUK-Grid – a new UK dataset of gridded climate observations, Geoscience Data Journal, submitted.

Met Office: HadUK-Grid gridded and regional average climate observations for the UK, Centre for Environmental Data Analysis, 10.5285/4dc8450d889a491ebb20e724debe2dfb, 2018.

Met Office: MIDAS Open: UK daily rainfall data, v201901, Centre for Environmental Data Analysis, 10.5285/ec54d5e5288a4ebb8c7ad2a1ef6aec42, 2019.

Smith, K.A.; Tanguy, M.; Hannaford, J.; Prudhomme, C. (2018). Historic reconstructions of daily river flow for 303 UK catchments (1891-2015). NERC Environmental Information Data Centre. https://doi.org/10.5285/f710bed1-e564-47bf-b82c-4c2a2fe2810e

Smith, K., Barker, L., Tanguy, M., Parry, S., Harrigan, S., Legg, T., Prudhomme, C., and Hannaford, J.: A Multi-Objective Ensemble Approach to Hydrological Modelling in the UK: An Application to Historic Drought Reconstruction, Hydrology and Earth System Sciences Discussions, https://doi.org/10.5194/hess-2019-3, 2019.

Svensson, C., Hannaford, J., and Prosdocimi, I.: Statistical distributions for monthly aggregations of precipitation and streamflow in drought indicator applications, Water Resources Research, 53, 999-1018, 2017.

---

## Author Comment (AC3) · 5 Aug 2019

**Gerry Spraggs - short comment
Author Response**

An excellent and interesting paper and good to see a national scale analysis of historic drought. I have some general comments which I hope may be helpful:

**Gerry, thank you for kind words and we are glad you found our paper on historic**

[Figure]

**hydrological droughts interesting. We have responded to each comment in turn below in bold.**

In the abstract you could say '108 near natural catchments'. I think it would be useful for water managers to know up-front the type of catchment you are analysing as they deal with a whole range of catchment conditions..

**Agreed, in the abstract of the revised paper we will include that this study uses near-natural catchments (P1L19).**

page6 lines9-16 To expand on the previous comment, much of the time water resource planning has to deal with non-natural flows. For this situation your method would require naturalisation of recorded flow series before model calibration and flow series extension. Alternatively, model calibration with artificial influences included. It's obviously achievable but a longer procedure with more room for introducing error.

**This is certainly a good point, and the modelling approach used by Smith et al. (2018, 2019) does not explicitly account for human influences. However, it does a reasonable job at modelling influenced flows. However, here we assess drought severity in near natural catchments, i.e. the natural variability, and as we allude to in the Discussion P25L27-31 further analysis is needed, however, in the revised version of the manuscript we will mention the need for inclusions of human influences (as well as groundwater) to assess the impact of droughts on water resources.**

page6 line23 Would it be worth appending a list in the supplement of the 108 catchments shown in Figure 1? I found myself trying to work out which Anglian catchments were used and others may want to do the same in other parts of the UK..

If the editor feels it is appropriate we could add a list of catchments to the supplementary information. Additionally, there is a list of catchments in the metadata of Barker et al. (2018) which includes a flag as to whether they are in the Low Flow Benchmark Network, however this is the full 115 catchments, and without cross referencing with Smith et al. (2018) it would not be possible to establish which 7 catchments were excluded from this study (as they didn't meet the highest performance criteria as described by Smith et al. (2019)).

Figure 2. The Maximum Intensity appears be dimensioned in units of time. Should it be defined by say a horizontal dotted line from the lowest SSI point (-2) to the Y axis, plus a vertical line (with arrowheads at each end) from the dotted line to SSI zero line? The dimension would then be SSI.

**We will adjust this figure in the revised paper to make it clear Maximum Intensity refers to the lowest SSI value within an event and does not have a time dimension.**

page14 lines7-8 1989-90 was severe in East Anglia, particularly for groundwater. So are you saying '...with the latter being particularly severe for the 1990s as a whole'?

**We will clarify this in the revised paper – we meant to say that in Anglian Region events ranked highly (and were therefore severe) in the 1990s. When looking at Fig S3b, events in the early half of the 1990s and late 1990s rank highly in terms of accumulated deficit.**

Figure 6. Looking at Anglian region it is clear that use of different characteristics identifies different droughts i.e.1891-1910, 1920-22, 1975-6, 1990-92. Water resource modelling often shows that there is sometimes very little difference between major droughts when it comes to defining system yield for use in the supply-demand balance. If you

look at Fig. 10 of our 2015 paper you can see that simulated reservoir drawdown at Grafham was very similar for 4 droughts: 1920-1, 1933-4, 1944-7 and 1975-6. Tweaks to the WR model parameters, e.g. frequency of supply restrictions, have been shown to invoke one or other of these droughts as critical. The point I'm making is that only by simulating a WR system over the whole historic series (behavioural analysis) will the critical drought be found - I think you say this later in the paper!

**Here we were concerned with which hydrological droughts were the most severe without the effect of management and without considering their impacts on water resources, society or the environment etc. We feel this is an important first step before impacts can be fully assessed. As you point out, we make this point later in the paper (P25L15-31). It would be an interesting next step to run the reconstructions through supply system models to assess the impact of these droughts on water supplies at the national scale.**

page20 line11 better to stick with 'near natural catchments'?

**Agreed, we will change this in the revised paper.**

page22 lines5-6 we found 1989-92 to be the most severe in the north of the Anglian region, including the Lud catchment, not the whole region (2015 paper Table 2). It ranked only 9th for Alton in Suffolk. But as you point out: different method, different durations.

**Apologies for this generalisation, we removed some of the detail at the last minute and we lost this clarification, we will make it clear this refers to the Lud only in the revised paper (which is the only catchment which overlaps between the two studies).**

page23 lines 6-10 1943-46 drought significant in west of Anglian region (2015 paper abstract and Table 2 for Grafham 24 months period)

**In the revised paper we will add that 1943-1946 was identified as significant by Spraggs et al. (2015) here.**

page24 lines12-13 Extending the hydrological series back from 1920 to 1800 did not introduce different critical droughts - they all remained post-1920 (2015 paper Table 3). It didn't change the approach or methodology, so could you delete 'approaches' and just say '...planning in particular water...'?

**Thank you for pointing this out, we will remove 'approaches' in the revised manuscript.**

page 24 lines17-21 Totally agree! I noted this in our 2015 paper Conclusions point 6.

**Indeed, we can add a reference to Spraggs et al. (2015) here in the revised paper.**

page 25 lines15-22 'non-stationarities in catchment response or land use change' may not be an issue for water resource planning. Current or projected future (planned) artificial catchment influences can be added to an extended naturalised series for use in water resource models. Catchment change etc. would of course be important for corroboration with documentary evidence.

**We appreciate that historic changes in catchment response or land use change may not an issue for water resources planning, but as we state here, modelling approaches do not account for changes in land use etc. over time and as such it is an important caveat to make for the reconstructed flows as they may not fully represent the past. It is also an important reason as to why the maintenance of long-term records and the digitisation and rescue of data are of critical**

**importance for hydrology.**

The first referee questions why the choice of 3 and 12 month SSI. From the water resources planning and management perspective longer droughts have been a concern, notably during the 2010-12 episode, with the ever increasing impact of global warming. So, although drought structure under a changing climate is conjectural, 24 and 36 month SSI would be interesting!

**We will note the importance of longer accumulation periods for water resources (particularly in the south-east of England) in the discussion of the revised paper. We will also note that results and data for additional accumulation periods can be viewed in the UK Hydrological Drought Explorer (https://shiny-apps.ceh.ac.uk/hydro_drought_explorer/) in the revised paper on P24L25.**

And a few typos:

**Thank you for spotting these typos, we will correct them in the revised manuscript.**

page2 line34 'quantify and understand'?
page8 line12 delete the ', they'
page 12 line20 you use the word 'record' when technically the earlier data is not recorded, so perhaps say 'period'? Anywhere else in paper?
page16 line14 'major droughts for'?
page17 line18 1890-1910
page17 line 21 'regularly'
page17 line29 At
page23 line19 Should it be Figure 5 or 7?

Kind regards, Gerry Spraggs

**Thank you again for your very helpful comments Gerry, we will work these into the revised version of the manuscript.**

**References**

Barker, L. J., Smith, K. A., Svensson, C., Tanguy, M., and Hannaford, J.: Historic Standardised 30 Streamflow Index (SSI) using Tweedie distribution with standard period 1961-2010 for 303 UK catchments (1891-2015), NERC Environmental Information Data Centre, 10.5285/58ef13a9-539f-46e5- 88ad-c89274191ff9, 2018.

Smith, K.A.; Tanguy, M.; Hannaford, J.; Prudhomme, C. (2018). Historic reconstructions of daily river flow for 303 UK catchments (1891-2015). NERC Environmental Information Data Centre. https://doi.org/10.5285/f710bed1-e564-47bf-b82c-4c2a2fe2810e

Smith, K., Barker, L., Tanguy, M., Parry, S., Harrigan, S., Legg, T., Prudhomme, C., and Hannaford, J.: A Multi-Objective Ensemble Approach to Hydrological Modelling in the UK: An Application to Historic Drought Reconstruction, Hydrology and Earth System Sciences Discussions, https://doi.org/10.5194/hess-2019-3, 2019.

Spraggs, G., Peaver, L., Jones, P., and Ede, P.: Re-construction of historic drought in the Anglian Region 20 (UK) over the period 1798–2010 and the implications for water resources and drought management, Journal of Hydrology, 526, 231-252, https://doi.org/10.1016/j.jhydrol.2015.01.015, 2015.

---

## Author Response (AR1)

Dear Editor,

Re: Submission of revised manuscript 'Historic hydrological droughts 1891-2015: systematic characterisation for a diverse set of catchments across the UK' hess-2019-202 for HESS.

Thank you for the opportunity to revise our manuscript, especially the additional time in which to do so. We appreciate your
5   comments as well as both reviewer comments and have responded to them below in italics, changes in the manuscript are marked with track changes below.

Yours sincerely,

Lucy Barker

**Response to editor comments**

Please replace as suggested Legg and McCarthy with the MetOffice references. Please also make sure that you refer to the newly published Smith et al (2019) paper where appropriate (e.g. the drought uncertainty consideration mentioned at the end of comment D) and that you include a short discussion of the uncertainties involved in your drought analyses.

5 *We have removed the Legg & McCarthy (in preparation) reference and replaced it with references to the published datasets and also added the reference Hollis et al. (2019) which has just been published and describes the data rescue activities. We have added references to Smith et al. (2019) where appropriate. We have also added a short discussion around the uncertainties of using the SSI for these analyses in Section 4.3.*

Also, I recommend adding as a supplement a list of the 108 catchments used in the study, with the most important metadata

10 (river/gauge name, size, hydrolclimatic region, etc.) plus, if helpful, a reference to Barker et el (2018).

*As you, and Gerry Spraggs, suggested we have added a list of the catchments and their properties to the Supplementary Information.*

**Response to reviewer 1**

**Major Comments**

15     A) The paper has a considerable inconsistency in terms of citation style. Please check all the citations to make sure that e.g. Authors et al. (2019), (Authors et al., 2019) and so on is used in a consistent way. This will improve the readability of the paper! Some examples are listed in the technical comments.

    *We have checked the citation style throughout and corrected where necessary.*

    B) The reference Legg and McCarthy (prep.) (P05L09) is really problematic for me. As the readers have no chance to access this paper and "preparation" is for me different to "is submitted", the authors should at least give a short description of what is done in the Legg and McCarthy paper. After all, the model is fed by this data and therefore it is important to understand how meteorological data there is "rescued and digitized". The same is partly true for

25     Smith et al. (2019) as this paper is still under review, isn't it? I suggest to give the reader whenever possible at least a brief description of data/method etc. instead of referring to unpublished studies. I can understand that this is not always easy to do, but it seems to be important to give the reader the chance to understand what has been done. It is also hard to understand how well the model performed (P6L18-L24) in detail, as no further information is given: Here my question is, how valuable is the modelling regarding low flows and streamflow droughts? Here more

30     justification is needed.

    *We have removed all citations of in preparation papers. Smith et al. 2019 has been revised and accepted in HESS and the reference updated. The Legg & McCarthy reference has been replaced with references to the published datasets and the recently published Hollis et al. (2019).*

    C) Regarding the model GR4J I have some concerns regarding the details of the modelling approach. The 4-parameter version is used, if I understand the details in the give references correctly. From Smith et al. (2018) I cannot learn much about the 4 parameters and the functioning, Smith et al. (2019) certainly gives more information on the parameters, but how do you justify that this modelling approach is appropriate for your study propose (i.e. non-

40     stationarity, long series, appropriate for low flows in different seasons)? Especially the slow component and its model representation is of great interest, as the slowest (groundwater) box in the model and its parameterization have potentially a high impact on drought characteristics (such as intensity, duration, deficit). Please comment on this issue (i.e. parameter sensitivity). Are there studies proofing that GR4J is a valuable modeling approach for low flow and drought analysis? Excluding snow and snowmelt processes might be reasonable, but that means that these

45     processes are not relevant for low flows and streamflow droughts in none of the study catchments?

*We have added more detail on this in Section 2.1 of the paper, including a reference to Caillouet et al (2017) who used the GR4J model to reconstruct low flows in France. Harrigan et al is also cited as they demonstrated good performance of the model across the UK. In response to other reviewer comments, we have added the list of metrics used by Smith et al in the calibration of the model used in this study, which include two low flows specific metrics: APE of Q95 and APE of Mean Annual Minimum on a 30 day accumulation period. Smith et al. (2019) conducted extensive parameter uncertainty estimation on the model, so we refer you to that paper for more detail. The Supplementary Info now includes plots of the performance of the model according to these metrics (Fig S1), and though the performance is varied across the country, we are satisfied with the results. The histograms below show that the majority of the LFBN catchments had low absolute percent error in MAM30 and Q95.*

[Figure]

D) A provocative comment: You stated that historical droughts have been more severe than recent droughts (i.e. observed droughts) and a historical assessment is important to better understand the potential drought magnitude in a region/country. Contrary to that, I would argue that the use of water is adjusted to the water availability of the last, let's say, max. 30-40 years. All water users can only use available water and changes in water availability on a time scale of 3-4 decades influences (of course!) the water uses/water users. So, why is The Long Drought at the beginning of the last century relevant for the water users today? If you show these nice heatmaps with drought severity over 125 years you should also show a heatmap of uncertainty (i.e. comparison between observation period after 1950s and model period before 1950s) (cf. P25L05). Here, I speculate that the uncertainty assessment will soften your statements about historical drought magnitude, duration, intensity.

*We appreciate your point that drought events may not have the same impacts now as they have done previously due to more resilient water supply and management systems. But regardless of water use, water resource managers look at natural water availability in their drought management plans. In the past, UK water supply drought plans have been based around planning for the worst event on record, and water companies must now plan for events outside of the historic record. Critical to these approaches is an understanding of events that have occurred in the past. Here we have identified past instances of events where natural water availability has been significantly lower, and for longer time periods than we have experienced in the recent past. Despite adjustments in water use to availability, extreme water deficits will still impact society, so information to better inform water resource managers on the characteristics of such events will always be valuable. The additional data provided by the reconstructed flow data provide this long view and enable the consistent identification and characterisation of droughts over the past 125 years. However, the uncertainty resulting in using only LHS1 (i.e. the best model run for each catchment as identified*

*by Smith et al. 2018) has been highlighted in Section 4.3. From a set of nine case study catchments, Smith et al (2019) found that parameter uncertainty had some impact on the extracted drought events, but mostly in the timing of droughts rather than the magnitude.*

**Minor comments**

- P02L05-10: How is the statement "historical records are still of fundamental importance in drought planning" justified? From my perspective Brown et al. highlights the lack of historical analysis, but the authors also referred to other studies in paper. However, I suggest to strengthen the study motivation here with more details on the value of historical data or analysis.
  *As discussed in the response to the reviewer, this section of the introduction was intended to introduce the benefits of using of historic data in planning approaches; later in the introduction for example on pages 3 and 4, the motivation of this study is more clearly defined.*

- P06L17-20: Would be helpful to give some more information about the criteria used to evaluate the performance.
  *As discussed in the response to the reviewer, we have continued to direct the reader to Smith et al, 2019, but have listed the six evaluation metrics used in Section 2.1 of the paper.*

- P06L26-30: What is the justification to select particularly these nine case study catchments? It is also not clear why case study catchments are used?
  *As discussed in the response to the reviewer, the case study catchments were selected in order for results to be shown for individual catchments and were chosen to represent a range of catchment sizes/characteristics with one catchment per region as stated on P6L34-P7L3.*

- P02L11: Just a suggestion: Are there some reference studies that have investigated major, severe droughts in UK? Could the paragraph better be linked to the P03L15-25) where some historical investigations have been listed?
  *As discussed in the response the reviewer, Marsh et al. 20007 has been added to this sentence as an exemplar reference.*

- P02L20: Is it warm/dry or warm and dry weather?
  *This has been change to 'warm and dry weather'.*

- P07L04: "end-month"? Is this the same as "right-aligned"?
  *As discussed in the response to the reviewer, this has been clarified in the text with an example for the three and twelve month accumulation periods in the revised text.*

- Sect 2.2.: I get the idea to have a short- and a long-term analysis (3 and 12 months). However, have you tested other accumulation periods? Is 12 month long enough to capture also long-term anomalies in the slowly reacting, GW dominated systems in South East England? As events with "less than three months were removed" (is this <3 month or <=3month?), I wonder why the SSI3 is used (as also a "seasonal focus" of the study is stated (P07L29) (see also comment below).
  *As discussed in the response to the reviewer, we have clarified that results for additional accumulation periods are available via the UK Hydrological Drought Explorer on P26 L28-29. We also clarified that events of 1 and 2 months were removed on P8 L13-14. The SSI-3 was taken to be analogous to seasonal deficits as UK seasons are generally determined to be around three months long. SSI-12 was selected as it encompasses deficits over multiple seasons, representing longer term deficits as is stated on P7 L13-22, in the revised manuscript we added a comment to say that it may be appropriate to use other accumulation periods (including longer ones) in some cases – particularly in the south-east of England.*

- What means "broadly north to south" exactly (P09L04)? Have you tried the heatmap with squares instead of rectangles (and with a fine border/stroke around the squares; this could improve the clarity of the graph, perhaps.). It would be also interesting to sort the catchments within each geographical group. North-to south is perhaps not really hydrological meaningful; what about a sorting along a low flow metric (e.g. Q90/Q50) to highlight differences in on-set and termination?

*We carefully considered the recommendation of reviewers 1 and 2 to sort the catchments of the y-axis of Figure3/S1 etc.by Q90. However, we felt that the arrangement of the catchments from north to south using the NRFA station numbers to sort catchments (which as described in the response to the reviewer means they are arranged by hydrometric area, and therefore grouped by climatologically and hydrologically similar areas – see here for more info: https://nrfa.ceh.ac.uk/station-number and https://nrfa.ceh.ac.uk/hydrometric-areas) better reflects the aims of the paper in understanding when droughts occurred across the UK, and when and where they were most severe. We also did not want to introduce another metric (Q90) which is not used elsewhere in the paper that is a different in concept to the Standardised Streamflow Index used. We have therefore integrated the other suggestions made by the reviewer around moving the region labels to outside of the plotting area etc. but have left the catchments ordered as they were previously.*

- Fig.4: Are the differences between maximum intensity (dot size) and mean deficit (colors) discussed?

*These characteristics are described on P12L4-7.*

- I am not an expert for historical droughts in UK, but is "The Long Drought" really a 20 year event without drought termination / interruptions? From Fig. 3 and Fig 10a, I have the impression that there are also a lot of "yellow" and "white" segments in the heatmap (e.g. 1904 wasn't really a dry year).

*This point was also raised by reviewer 2. We have amended any reference to the 'Long Drought' to "the 'Long Drought' period" to reflect that it was a period within which drought conditions occurred (rather than a continuous period of drought) as is shown in Figures 3 and 10a.*

- Fig. 6 is really a nice idea, but it is hard to understand and it take me a while to understand the encodings used in the Figure. I suggest to use a UK-matrix with 9 columns (i.e. events) and 4 rows (i.e. drought characteristics). Then in each subplot all catchments with mild grey dots overplotted by the top ranking catchments in black color. Would improve the clarity of the Fig.

*We have added an additional key to Figure 6, and subsequent plots in the same style, to illustrate which characteristic each colour and circle style represents. We think this has made these figures easier to interpret.*

- Would be interesting to quantify the differences between the MCW2007 drought magnitude and the (more severe) droughts on catchment or regional scale (Sect 4.1), e.g. what is the difference of a very critical drought situation in a specific catchment compared to the "national" drought magnitude?

*As we discuss in the response to the reviewer, the focus of this paper was the consistent identification, characterisation and ranking of events at the national scale. We have highlighted on P27L34 that future work should assess drought severity in a more holistic way, by including rainfall, groundwater and water supply analyses, at the national and regional scale.*

- The authors stated that SSI-3 and SSI-12 are a good choice to identify different drought types (P23). Is this a general recommendation for other studies (3- and 12-months)? If not, what might be a good (and sufficient) set of different SSI-n to capture the variability of historical droughts?

*We use the 3 and 12 month accumulation period to characterise single season (3-month) and multi-season annual (SSI-12) hydrological droughts as described on P7 L13-16, and would recommend these accumulation periods for these purposes. However, the exact choice of accumulation period in future studies will depend on the motivation and application of research, if for example, you are interested in multi-year droughts you may choose to look at*

*accumulation periods of 12, 24, 36 months etc.) – we have added a comment to this effect on P7 L16-22. However, we felt that the use and presentation of additional accumulation periods was out of the scope of this paper.*

- Sect 4.3 is a little bit long and could be more condensed. The authors discussed potential limitations of their work (e.g. non-stationarity, model uncertainty), but here I missed a clear link to the (own) study results.
  *We have condensed Section 4.3 in the revised paper whilst introducing other points recommended by the other reviewers.*

**Technical Comments**

1. P06L05: Smith et al. (2019) also assessed
   *Citation style updated.*

2. P06L09: by Smith et al. (2018)
   *Citation style updated.*

3. P06L11: Low Flow Benchmark Network (LFBN).
   *Low Flow capitalised.*

4. P06L17: reconstructed by Smith et al. (2018), which include the LFBN, performed
   *Text updated as suggested.*

5. For readers from outside UK a short explanation of "Anglian" would be helpful (P09L23).
   *Anglian changed to ANG to make clear it refers to the hydroclimate region shown in Figure 1. Updated all mentions of regions to the acronyms used in Figure 1 for clarity.*

6. P11L03-04: two times "accumulation period"?
   *Grammar of sentence improved and two mentions of accumulation periods removed.*

7. lower maximum intensity is more severe? (P11L04/05). Terms should be revised here.
   *Clarified that lower maximum intensity is a more severe event.*

8. Fig.4: The 45 degree axis labels are hard to read, thin grid lines or a lollipop graph instead of bubble graph could improve the readability. If you referred to pre-obs and obs-period than a vertical line to distinguish both periods would be beneficial. Have you tried a lollipop chart here, i.e. vertical lines between dots and x-axis might improve the readability?
   *Plots amended so that x-axis labels are at 90° and so there is a vertical line between the points and the x-axis. We did not add the vertical line to mark the pre-observation/observation period but have added a comment to the text to make clear when we are referring to.*

9. Remove leading white spaces in (*Figure 5. . .) on page 12.
   *Formatting issue resolved.*

**Response to reviewer 2**

**General comments:**

Little information is given on the basic datasets used for driving the hydrological model. Please elaborate in more detail on the digitized meteorological data. Is this raw data or have they undergone a homogenization procedure? I also think that a reference for a paper in "preparation" is not suitable. Moreover I think that there has to be a more in-depth description of the hydrological

modelling. E.g. Smith et al. (2019) used six evaluation metrics some of the specific for low flows. What are these metrics and what is the performance? Please provide some information in this respect.

*We have removed the reference to the in preparation Legg & McCarthy reference and replaced it with references to the published datasets as stated in the response to the reviewer and a newly published paper Hollis et al. (2019) which describes the digitisation process of the data rescue and the datasets.*

*We have added a list of the model performance metrics used by Smith et al. (2019) to assess model performance, and have added a figure to the supplementary information (Figure S1) which maps the six LHS1 model performance metrics for the 108 LFBN catchments.*

You use the SSI as a standardized hydrological drought indicator. What about the uncertainties considering the fitting of the distribution and how do these translate in terms of derived drought metrics? Since you use mostly rankings of the top events it is rather crucial how the fitting performs particularly at the tails of the distribution. Could you just exemplarily give an indication of possible change in the ranking of some drought metric from fitting uncertainty?

*As discussed in the response to the reviewer, we have added more information on the benefits of using the Tweedie distribution to the revised paper on P7 L24-28, and a brief discussion of the uncertainties to the discussion on P26 L33-P27 L1.*

Figure layout:

For Figures 3, 5 and 10 I suggest to place the acronyms for the region outside the plot area along the y-axis for better readability. Also rethink the arrangement of catchments along the y-axis, perhaps there is a better way than a strict North/South (driven by climate) alignment (e.g. low flow characteristics).

*We have moved the region labels to the y-axis, making the plot easier to read.*

*We carefully considered the recommendation of reviewers 1 and 2 to sort the catchments of the y-axis of Figure3/S1 etc.by Q90. However, we felt that the arrangement of the catchments from north to south using the NRFA station numbers to sort catchments (which as described in the response to the reviewer means they are arranged my hydrometric area, and therefore grouped by climatologically and hydrologically similar areas – see here for more info: https://nrfa.ceh.ac.uk/station-number and https://nrfa.ceh.ac.uk/hydrometric-areas) better reflects the aims of the paper in understanding when droughts occurred across the UK, and when and where they were most severe. We also did not want to introduce another metric which is not used elsewhere in the paper that is a different in concept to the standardised Streamflow Index used. We have therefore made the other suggestions around moving the region labels to outside of the plotting area etc. but left the catchments ordered as they were previously.*

Figure 5: The colorbar as a gradient from red to yellow is in general appropriate for this kind of data in terms of figure layout guidelines. However, since the displayed data is a ranking, I think that the reader would like to see first of all where the top ranked events are. This is not easy in this case. Perhaps you could try a colorbar with more colors? (in R: RColorBrewer palette "Spectral") Or combine two colorbars, one for the top 3 (or 5?) and one for the rest. Don't know how it would look, but it is perhaps worth a try to get the essential information better across.

*We have added a dark purple to the colour palette used (as the colour for rank 1), and removed one of the red colours. We feel this makes it easier to separate the different ranks and the plot easier to interpret. Palettes with too many colours, such as "spectral", are not suitable for colour-blind readers.*

Several times across the manuscript I stumbled over the terms droughts, drought event or drought periods. I'd like to see more consistency with these terms. The list of major droughts (Table 1) is mostly termed events, however, the 1890-1910 period is not an event from an event definition point of view. This comes rather clear in Figure 10a, where the "long drought" is clearly made up of several individual events(!) all of them with a distinct beginning and end. On the other hand, 1921 (Figure 10b) is clearly an event itself, it has a distinct beginning and end. I suggest to define the names of the major droughts as in Table 1 and stick to the terms, e.g. "1890-1910 drought period", "1921 event", "1976 event", etc. I think that an event stretching over

several years could be termed as the "year xxxx event", with the year being that with maximum drought intensity for example, which has to be defined obviously.

*We have referred to the Long Drought as "the 'Long Drought' period" throughout, we hope that this reflects that is a name given to the period as a whole. We have continued to refer to the remaining events as start year – end year as discussed in the response to the reviewer.*

**Specific comments:**

P2L20: ". . .short periods of warm and dry weather. . ."

*This has been changed as suggested.* P2L24: "Moreover, greater climatic variability could mean an increase in persistent blocking episodes and multi-year droughts" please provide a reference for this statement.

*We have cited Folland et al. 2015 in this sentence.*

P6L5: "Smith et al. (2019) also" please be generally careful with the citations, there are some other inconsistencies.

*The citation style here has been corrected and remaining citations checked and corrected where necessary.*

P6L11: "Low Flow Benchmark Network (LFBN)"

*Low Flow has been capitalised as suggested.*

P11L3: suggestion: "For both time scales considered, events tend. . ."

*We have modified this sentence to improve the grammar.*

P19L6: ". . . e.g. 1895 saw extreme flow deficits across Scotland and Northern Ireland. . ."

*This sentence has been changed as suggested.*

P22L10: In this section some recent research would be appropriate to cite, since there are some events detected in the present paper also listed as extreme droughts in other regions of Europe for example in:

Hanel, M., Rakovec, O., Markonis, Y., Máca, P., Samaniego, L., Kyselá, J., & Kumar, R. (2018). Revisiting the recent European droughts from a long-term perspective. Scientific Reports, 8(1). https://doi.org/10.1038/s41598-018-27464-4

Haslinger, K., & Blöschl, G. (2017). Space-Time Patterns of Meteorological Drought Events in the European Greater Alpine Region Over the Past 210 Years. Water Resources Research, 53(11), 9807–9823. https://doi.org/10.1002/2017WR020797

*We have added reference to Hanel et al. (2018) to Section 4.1 of the revised manuscript, we felt that Haslinger et al. (2017), although had similar and related results was too specific in spatial coverage, and so on reflection did not add this reference to the revised manuscript.*

**Response to Gerry Spraggs**

In the abstract you could say '108 near natural catchments'. I think it would be useful for water managers to know up-front the type of catchment you are analysing as they deal with a whole range of catchment conditions.

*We have added that the 108 catchments were near-natural to the abstract.*

page6 lines9-16 To expand on the previous comment, much of the time water resource planning has to deal with non-natural flows. For this situation your method would require naturalisation of recorded flow series before model calibration and flow series extension. Alternatively, model calibration with artificial influences included. It's obviously achievable but a longer procedure with more room for introducing error.

*In response to the comments of the other reviewers, Section 4.3 was condensed and so this comment was not added in the end, although clearly will be an extremely important consideration of any further work.*

page6 line23 Would it be worth appending a list in the supplement of the 108 catchments shown in Figure 1? I found myself trying to work out which Anglian catchments were used and others may want to do the same in other parts of the UK.

*A table of the 108 catchments used here has been added to the Supplementary Information (Table S1).*

Figure 2. The Maximum Intensity appears be dimensioned in units of time. Should it be defined by say a horizontal dotted line from the lowest SSI point (-2) to the Y axis, plus a vertical line (with arrowheads at each end) from the dotted line to SSI zero line? The dimension would then be SSI.

*We have amended the plot as suggested.*

page14 lines7-8 1989-90 was severe in East Anglia, particularly for groundwater. So are you saying '...with the latter being particularly severe for the 1990s as a whole'?

*This has been clarified in the revised paper as discussed in the response to the reviewer.*

Figure 6. Looking at Anglian region it is clear that use of different characteristics identifies different droughts i.e.1891-1910, 1920-22, 1975-6, 1990-92. Water resource modelling often shows that there is sometimes very little difference between major droughts when it comes to defining system yield for use in the supply-demand balance. If you look at Fig. 10 of our 2015 paper you can see that simulated reservoir drawdown at Grafham was very similar for 4 droughts: 1920-1, 1933-4, 1944-7 and 1975-6. Tweaks to the WR model parameters, e.g. frequency of supply restrictions, have been shown to invoke one or other of these droughts as critical. The point I'm making is that only by simulating a WR system over the whole historic series (behavioural analysis) will the critical drought be found - I think you say this later in the paper!

*As stated in the response to the reviewer, here we were concerned with which hydrological droughts were the most severe without the effect of management and without considering their impacts on water resources, society or the environment etc. We feel this is an important first step before impacts can be fully assessed. As you point out, we make this point later in the paper (P27 L34- P28 L4). It would be an interesting next step to run the reconstructions through supply system models to assess the impact of these droughts on water supplies at the national scale.*

page20 line11 better to stick with 'near natural catchments'?

*Changed as suggested.*

page22 lines5-6 we found 1989-92 to be the most severe in the north of the Anglian region, including the Lud catchment, not the whole region (2015 paper Table 2). It ranked only 9th for Alton in Suffolk. But as you point out: different method, different durations.

*We have clarified that this statement referred only to the River Lud as discussed in the response to the reviewer.*

page23 lines 6-10 1943-46 drought significant in west of Anglian region (2015 paper abstract and Table 2 for Grafham 24 months period)

*We have added to the revised paper that Spraggs et al. 2015 also found this event to be significant for the Anglian region.*

page24 lines12-13 Extending the hydrological series back from 1920 to 1800 did not introduce different critical droughts - they all remained post-1920 (2015 paper Table 3). It didn't change the approach or methodology, so could you delete 'approaches' and just say '...planning in particular water...'?

*Changed as suggested.*

page 24 lines17-21 Totally agree! I noted this in our 2015 paper Conclusions point 6.

*We appreciate that this was noted by Spraggs et al. (2015) but on reflection we felt it was appropriate to reference this paper in this sentence, we have however, noted in many other places the synergies between the two papers.*

page 25 lines15-22 'non-stationarities in catchment response or land use change' may not be an issue for water resource planning. Current or projected future (planned) artificial catchment influences can be added to an extended naturalised series for use in water resource models. Catchment change etc. would of course be important for corroboration with documentary evidence.

*As noted in the response to the reviewer, we appreciate that historic changes in catchment response or land use change may not an issue for water resources planning, but as we state in the paper, modelling approaches do not account for changes in land use etc. over time and as such it is an important caveat to make for the reconstructed flows as they may not fully represent*

*the past. It is also an important reason as to why the maintenance of long-term records and the digitisation and rescue of data*
*are of critical importance for hydrology.*

The first referee questions why the choice of 3 and 12 month SSI. From the water resources planning and management perspective longer droughts have been a concern, notably during the 2010-12 episode, with the ever increasing impact of global
5 warming. So, although drought structure under a changing climate is conjectural, 24 and 36 month SSI would be interesting!
*We have added a sentence to describe the necessity of assessing additional accumulation periods based on the location and*
*the event of interest to P7L17-19 We have also noted that results for additional accumulation periods can be viewed on the*
*UK Hydrological Drought Explorer (https://shiny-apps.ceh.ac.uk/hydro_drought_explorer/) on P26 L27.*

And a few typos:
10 page2 line34 'quantify and understand'?
*Changed as suggested.*
page8 line12 delete the ', they'
*Changed as suggested.*
page 12 line20 you use the word 'record' when technically the earlier data is not recorded, so perhaps say 'period'? Anywhere
15 else in paper?
*Changed to say earlier part of the reconstructed series.*
page16 line14 'major droughts for'?
*Changed as suggested.*
page17 line18 1890-1910
20 *Corrected.*
page17 line 21 'regularly'
*Corrected.*
page17 line29 At
*Corrected.*
25 page23 line19 Should it be Figure 5 or 7?
*Figure numbers corrected.*

[revised manuscript text omitted]

**Supplementary information**

**Table S1 The 108 LFBN catchments used in this study, their hydroclimate region and area (from: National River Flow Archive, 2019). For more information about catchments see the National River Flow Archive ([www.nrfa.ceh.ac.uk/](http://www.nrfa.ceh.ac.uk/)). The nine case study catchments are marked with an asterisk.**

| NRFA Station Number | Catchment Name | Hydroclimate Region | Area (km$^2$) |
|---:|---|:---:|---:|
| 3003 | Oykel at Easter Turnaig | WS | 330.7 |
| 7001 | Findhorn at Shenachie | WS | 415.6 |
| 8004 | Avon at Delnashaugh | ES | 542.8 |
| 8009 | Dulnain at Balnaan Bridge | ES | 272.2 |
| 12001 | Dee at Woodend | ES | 1370 |
| 12005 | Muick at Invermuick | ES | 110 |
| 16003 | Ruchill Water at Cultybraggan | ES | 99.5 |
| 17005 | Avon at Polmonthill | ES | 195.3 |
| *18001 | Allan Water at Kinbuck | ES | 161 |
| 20007 | Gifford Water at Lennoxlove | ES | 64 |
| 21017 | Ettrick Water at Brockhoperig | ES | 37.5 |
| 21024 | Jed Water at Jedburgh | ES | 139 |
| 22001 | Coquet at Morwick | NEE | 569.8 |
| 23004 | South Tyne at Haydon Bridge | NEE | 751.1 |
| 24004 | Bedburn Beck at Bedburn | NEE | 74.9 |
| 25006 | Greta at Rutherford Bridge | NEE | 86.1 |
| 26003 | Foston Beck at Foston Mill | NEE | 57.2 |
| *27035 | Aire at Kildwick Bridge | NEE | 282.3 |
| 27042 | Dove at Kirkby Mills | NEE | 59.2 |
| 27047 | Snaizeholme Beck at Low Houses | NEE | 10.2 |
| 27051 | Crimple at Burn Bridge | NEE | 8.1 |
| 27071 | Swale at Crakehill | NEE | 1363 |
| 27073 | Brompton Beck at Snainton Ings | NEE | 12.9 |
| 28046 | Dove at Izaak Walton | ST | 83 |
| 28072 | Greet at Southwell | ST | 46.2 |
| *29003 | Lud at Louth | ANG | 55.2 |
| 29009 | Ancholme at Toft Newton | ANG | 27.2 |
| 30004 | Lymn at Partney Mill | ANG | 61.6 |
| 30012 | Stainfield Beck at Cream Poke Farm | ANG | 37.4 |
| 30015 | Cringle Brook at Stoke Rochford | ANG | 50.5 |
| 32003 | Harpers Brook at Old Mill Bridge | ANG | 74.3 |
| 33018 | Tove at Cappenham Bridge | ANG | 138.1 |
| 33029 | Stringside at Whitebridge | ANG | 98.8 |
| 34011 | Wensum at Fakenham | ANG | 161.9 |
| 36003 | Box at Polstead | ANG | 53.9 |
| 37005 | Colne at Lexden | ANG | 238.2 |
| 38026 | Pincey Brook at Sheering Hall | SE | 54.6 |
| *39019 | Lambourn at Shaw | SE | 234.1 |
| 39020 | Coln at Bibury | SE | 106.7 |
| 39025 | Enborne at Brimpton | SE | 147.6 |
| 39028 | Dun at Hungerford | SE | 101.3 |
| 39034 | Evenlode at Cassington Mill | SE | 430 |
| 40011 | Great Stour at Horton | SE | 345 |

| NRFA Station Number | Catchment Name | Hydroclimate Region | Area (km²) |
|---|---|---|---|
| 41022 | Lod at Halfway Bridge | SE | 52 |
| 41025 | Loxwood Stream at Drungewick | SE | 91.6 |
| 41027 | Rother at Princes Marsh | SE | 37.2 |
| 41029 | Bull at Lealands | SE | 40.8 |
| 42003 | Lymington at Brockenhurst | SE | 98.9 |
| 42008 | Cheriton Stream at Sewards Bridge | SE | 75.1 |
| 43014 | East Avon at Upavon | SE | 85.78 |
| 44006 | Sydling Water at Sydling St Nicholas | SE | 12.4 |
| 45005 | Otter at Dotton | SWESW | 202.5 |
| 46005 | East Dart at Bellever | SWESW | 21.5 |
| 47009 | Tiddy at Tideford | SWESW | 37.2 |
| 48003 | Fal at Tregony | SWESW | 87 |
| 49004 | Gannel at Gwills | SWESW | 41 |
| 50002 | Torridge at Torrington | SWESW | 663 |
| 52010 | Brue at Lovington | SE | 135.2 |
| 52016 | Currypool Stream at Currypool Farm | SE | 15.7 |
| 53006 | Frome (Bristol) at Frenchay | SE | 148.9 |
| 53008 | Avon at Great Somerford | SE | 303 |
| 53009 | Wellow Brook at Wellow | SE | 72.6 |
| 53017 | Boyd at Bitton | SE | 47.9 |
| *54008 | Teme at Tenbury | ST | 1134.4 |
| 54018 | Rea Brook at Hookagate | ST | 178 |
| 54025 | Dulas at Rhos-y-pentref | ST | 52.7 |
| 54034 | Dowles Brook at Oak Cottage | ST | 40.8 |
| 55014 | Lugg at Byton | SWESW | 203.3 |
| 55016 | Ithon at Disserth | SWESW | 358 |
| 55026 | Wye at Ddol Farm | SWESW | 174 |
| 55029 | Monnow at Grosmont | SWESW | 354 |
| 56013 | Yscir at Pont-Ar-Yscir | SWESW | 62.8 |
| *57004 | Cynon at Abercynon | SWESW | 106 |
| 60002 | Cothi at Felin Mynachdy | SWESW | 297.8 |
| 60003 | Taf at Clog-y-Fran | SWESW | 217.3 |
| 62001 | Teifi at Glanteifi | SWESW | 893.6 |
| 65001 | Glaslyn at Beddgelert | NWENW | 68.6 |
| 65005 | Erch at Pencaenewydd | NWENW | 18.1 |
| 66004 | Wheeler at Bodfari | NWENW | 62.9 |
| 67018 | Dee at New Inn | NWENW | 53.9 |
| 68005 | Weaver at Audlem | NWENW | 207 |
| 72005 | Lune at Killington | NWENW | 219 |
| 72014 | Conder at Galgate | NWENW | 28.5 |
| 73005 | Kent at Sedgwick | NWENW | 209 |
| 73011 | Mint at Mint Bridge | NWENW | 65.8 |
| *75017 | Ellen at Bullgill | NWENW | 96 |
| 76014 | Eden at Kirkby Stephen | NWENW | 69.4 |
| 77004 | Kirtle Water at Mossknowe | WS | 72 |
| 78004 | Kinnel Water at Redhall | WS | 76.1 |
| 79002 | Nith at Friars Carse | WS | 799 |
| 79004 | Scar Water at Capenoch | WS | 142 |

| NRFA Station Number | Catchment Name | Hydroclimate Region | Area (km²) |
|---|---|---|---|
| *81002 | Cree at Newton Stewart | WS | 368 |
| 81004 | Bladnoch at Low Malzie | WS | 334 |
| 83006 | Ayr at Mainholm | WS | 574 |
| 83010 | Irvine at Newmilns | WS | 72.8 |
| 84022 | Duneaton at Maidencots | WS | 110.3 |
| 85003 | Falloch at Glen Falloch | WS | 80.3 |
| 90003 | Nevis at Claggan | WS | 69.2 |
| 93001 | Carron at New Kelso | WS | 137.8 |
| 94001 | Ewe at Poolewe | WS | 441.1 |
| 96002 | Naver at Apigill | WS | 477 |
| 201008 | Derg at Castlederg | NI | 335.4 |
| 202002 | Faughan at Drumahoe | NI | 273.1 |
| 203028 | Agivey at Whitehill | NI | 100.5 |
| *203042 | Crumlin at Cidercourt Bridge | NI | 55.3 |
| 204001 | Bush at Seneirl Bridge | NI | 299.2 |
| 205008 | Lagan at Drumiller | NI | 84.6 |
| 206001 | Clanrye at Mountmill Bridge | NI | 120.3 |

**Table S2 Catchments and months with missing SSI-3 values**

| Catchment | Months with missing SSI-3 values | Impact |
|---|---|---|
| **29003 Lud at Louth** | 2007-09 | No impact on the extracted drought events |
| **40011 Great Stour at Horton** | 1921-12 | Splits a drought event which without the missing value would be the longest (and most severe in terms of accumulated deficit) event in this catchment |
| **54034 Dowles Brook at Oak Cottage** | 2007-07; 2007-08 | No impact on the extracted drought events |
| **72014 Conder at Galgate** | 1907-07 | No impact on the extracted drought events |

[Figure]

Figure S1 Model performance metrics from Smith et al. (2018) for the 108 LFBN catchments used in this study. Darker colours indicate better model performance.

[Figure]

**Figure S2 Heatmap of SSI-3 for LFBN catchments (arranged roughly from north to south on the y-axis with one row per catchment and regions marked for clarity) from 1891 to 2015.**

[Figure]

**Figure S3 Extracted events from SSI-3 and their characteristics for the nine case study catchments, plotted at the midpoint of the event. The size of each point is proportional to the maximum intensity and the colour indicates the mean deficit.**

[Figure]

**Figure S4 Top 10 extracted events from SSI-3 using a threshold of -1.5 for each drought event characteristic. Catchments are arranged roughly from north to south on the y-axis with each row representing a catchment. Bars represent the top 10 events and are coloured according to the event rank; darker shades represent higher ranking (i.e. more severe) events.**

[Figure]

**Figure S5 Location and number of LFBN catchments where the top ranking SSI-3 event corresponds to major events (Table 1) for duration (dur), accumulated deficit (accDef), mean deficit (meanDef) and maximum intensity (maxInt). Each of the nine maps represents one of the major drought events listed in Table 1. Each point on the maps represents the location of the 108 LFBN catchments. Points are coloured pink where the particular event was ranked most severe according to maximum intensity for that catchment. Similarly, points are circled in purple, orange and turquoise to indicate catchments where the particular event was ranked most severe in terms of mean deficit, accumulated deficit and duration, respectively. The numbers in the top right of each map show the number of catchments ranked as most severe for each characteristic for that particular event.**

[Figure]

**Figure S6 Months when SSI-3 top ranked events occurred outside of the major events (shaded in grey) for the LFBN catchments and each event characteristic (a-d), and e) the location and number of catchments with other top ranking events for each event characteristic. Points are coloured as described in the caption for Figure S5.**

[Figure]

**Figure S7 Location and number of LFBN catchments where the top ranking SSI-3 events for each event characteristic occur in periods outside of the major drought events: a) 1928-129, b) 1937-1938, c)1940-1949, d) 1960-1966, e) 1968-1975 and f) 1984. Points are coloured as described in the caption for Figure S5.**

[Figure]

**Fig S8 Boxplots showing the ranks of all extracted SSI-3 events where they overlap with the major drought events (top panel for each event characteristic) and identified 'other' events (bottom panel for each event characteristic). Within each box, n refers to the total number of events (across the LFBN) identified that occurred within this period. As multiple events can occur within each given period in individual catchments, it is possible for the value of n to be greater than the number of catchments (i.e. 108).**

[Figure]

**Figure S9 Heat maps of reconstructed SSI-3 for LFBN catchments, arranged roughly from north to south with one row per catchment and regions marked for clarity for a) the 'Long Drought' period (1890s-1910s), b) 1921-1922, c) 1933-1935 and d) the 1940s.**